# Robust and Interpretable Adaptation of Equivariant Materials Foundation Models via Sparsity-promoting Fine-tuning

**Youngwoo Cho**[*]
KAIST
cyw314@kaist.ac.kr

**Seunghoon Yi**[*]
Seoul National Univ.
jaguar6182@snu.ac.kr

**Wooil Yang**
KIAS
yspacefirst@kias.re.kr

**Sungmo Kang**
KIAS
smkang@kias.re.kr

**Young-woo Son**
KIAS
hands@kias.re.kr

**Jaegul Choo**
KAIST
jchoo@kaist.ac.kr

**Joonseok Lee**[†]
Seoul National Univ.
joonseok@snu.ac.kr

**Soo Kyung Kim**[†]
Ewha Womans Univ.
sookim@ewha.ac.kr

**Hongkee Yoon**[†]
Kangwon National Univ.
hongkeeyoon@kangwon.ac.kr

## Abstract

Pre-trained materials foundation models, or machine learning interatomic potentials, leverage general physicochemical knowledge to effectively approximate potential energy surfaces. However, they often require domain-specific calibration due to physicochemical diversity as well as mismatches between practical computational settings and those used in constructing the pre-training data. To address this, we propose a sparsity-promoting fine-tuning method that selectively updates model parameters by exploiting the structural properties of E(3)-equivariant materials foundation models. On energy and force prediction tasks across molecular and crystalline benchmarks, our method matches or surpasses full fine-tuning and equivariant low-rank adaptation while updating only ∼3 % of parameters, and in some cases as little as ∼0.5 %. Beyond energy and force calibration, we further demonstrate task generalizability by applying our method to magnetic moment prediction and magnetism-aware total energy modeling. Finally, analysis of sparsity patterns reveals physically interpretable signatures, such as enhanced $d$-orbital contributions in transition metal systems. Overall, our results establish sparsity-promoting fine-tuning as a flexible and interpretable method for domain specialization of equivariant materials foundation models.

## 1 Introduction

Machine learning interatomic potentials (MLIPs, Blank et al. (1995)) aim to reproduce the potential energy surface (PES) of material structures, serving as efficient surrogates for density functional theory (DFT, Kohn & Sham (1965)). Modern MLIPs offer remarkable efficiency and accuracy in regimes that have been computationally challenging for traditional physics-based simulations with limited scalability. Inspired by foundation models in computer vision and natural language processing (Caron et al., 2021; Li et al., 2022; Radford et al., 2021), recent work has leveraged large-scale materials datasets to develop atomistic foundation models (Barroso-Luque et al., 2024; Batatia et al., 2025; Chen & Ong, 2022; Deng et al., 2023; Park et al., 2024), achieving impressive accuracy on multiple benchmarks (Levine et al., 2025; Riebesell et al., 2025).

Due to the extreme diversity of material systems, however, even a large-scale pre-training dataset cannot fully cover the complicated and subtle landscape of materials, under-representing elements and motifs, as well as diverse physicochemical variations (*e.g.*, pressure and temperature). Thus, a

---

[*]Equal contribution. [†]Corresponding authors.

direct application of a pre-trained foundation MLIP often fails to generalize to unseen conditions, requiring domain-specific calibration (fine-tuning) of the foundation model. Moreover, downstream applications may rely on different levels of theory or exchange-correlation functional settings than those used to construct the pre-training data, introducing systematic discrepancies.

Given this necessity, the question shifts to how such calibration should be performed. Full fine-tuning, which updates all model parameters, would be the most straightforward adaptation method. However, domain-specific datasets are typically small relative to the vast configurational and chemical space they should represent, making full fine-tuning highly prone to overfitting, aside from its computational and memory costs. Parameter-efficient fine-tuning methods such as Adapters (Houlsby et al., 2019) and low-rank adaptation (LoRA, Hu et al. (2022)) provide a natural alternative by introducing a small set of trainable parameters. Applying these methods to equivariant graph neural networks (GNNs), which are commonly used in *state-of-the-art* MLIPs, requires preserving equivariance constraints, *i.e.*, translational, rotational, and reflectional symmetries in 3D space. To this end, GeoAda (Zhao et al., 2025) and ELoRA (Wang et al., 2025) redesign Adapters and LoRA, respectively, to preserve these constraints.

These methods focus on *how* the parameters are updated, *e.g.*, via low-rank structures or regularized magnitudes. A complementary yet unexplored perspective is to instead control *which* parameters are updated, encouraging the model to selectively modify only those most relevant to the target domain while preserving the rest intact. In other words, rather than constraining how updates are parameterized, one can directly regularize the *number of impacted parameters*.

Sparsity-promoting adaptation offers a natural realization of this idea, while also addressing a long-standing goal in scientific machine learning (ML): enhancing model interpretability by reducing unnecessary degrees of freedom (Brunton et al., 2016; Hoefler et al., 2021). MLIPs offer unique possibilities in this regard, as their internal layers often encode physically meaningful basis functions such as spherical harmonics (Geiger & Smidt, 2022). By selectively updating the weights associated with these bases, one can precisely identify which channels are modified during fine-tuning and which remain unchanged, providing clear insight into the model's internal physical representations. However, sparsity-promoting approaches remain underexplored in the context of MLIPs, particularly regarding their application to fine-tuning.

To bridge this gap, we propose a *sparsity-promoting fine-tuning method* for equivariant MLIPs, which carefully maintains equivariance with selective parameter updating. Applied to MACE potentials (Batatia et al., 2022; 2025; Kovács et al., 2025) across molecular, crystalline, and magnetic benchmarks, our method matches or exceeds the accuracy of full fine-tuning while updating only 0.5-3 % of parameters. Beyond recalibrating energy and force predictions, it extends effectively to other material properties such as magnetic moments, demonstrating broader applicability of foundation MLIPs. Finally, analysis of sparsity patterns reveals which coefficients are selectively updated, yielding physical insights into the learned representations.

Our contributions are threefold— **(i) Sparse equivariant fine-tuning:** We introduce a fine-tuning method on top of equivariant MLIPs that unifies symmetry preservation with a sparsity-promoting approach, matching or surpassing full fine-tuning even with extremely sparse updates. **(ii) Flexible and accurate adaptation:** Across molecular, crystalline, and magnetic benchmarks, we demonstrate the versatility of our method in accurately predicting physical properties across diverse domains. **(iii) Physically interpretable sparsity:** Analysis of sparsity patterns reveals physically interpretable signatures of the physicochemical representations learned by fine-tuned MLIPs.

## 2    RELATED WORK

This section presents a brief survey of related work. The mathematical and physical background needed to understand our study is provided in Appendix A.

**Machine Learning Interatomic Potentials.** In the early stages of MLIP development, simple feed-forward networks were employed on limited target systems, using atomic descriptors to predict energy (Behler et al., 2008; Blank et al., 1995; Le & Raff, 2008; No et al., 1997; Smith et al., 2017). Later, GNNs enabled sophisticated modeling of chemical interactions through message passing by representing atoms as nodes and bonds as edges (Gilmer et al., 2017; Schütt et al., 2017; Unke &

Meuwly, 2019). However, these models handle only invariant representations, without considering directional information crucial for vector and tensor quantities (*e.g.*, forces or dipoles).

Recognizing this limitation, architectures that preserve geometric symmetries have been developed, with features and outputs transforming under $SE(3)$ or $E(3)$ group operations. These transformations rely on type-$\ell$ vectors and spherical harmonics, combined with Clebsch-Gordan tensor products, inspired by physical principles (Geiger & Smidt, 2022; Kondor et al., 2018; Satorras et al., 2021; Thomas et al., 2018). Building upon these foundations, representative equivariant architectures such as MACE (Batatia et al., 2022), NequIP (Batzner et al., 2022), Allegro (Musaelian et al., 2023), and Equiformer (Liao & Smidt, 2023) are capable of capturing many-body interactions while strictly preserving physical symmetries.

These advances have further enabled the emergence of foundation MLIPs, following trends in vision and language domains (Caron et al., 2021; Li et al., 2022; Radford et al., 2021). Such foundation models are pre-trained on large-scale databases of DFT calculations (Barroso-Luque et al., 2024; Deng et al., 2023; Jain et al., 2013; Levine et al., 2025) to achieve broader generalization. Notable examples include M3GNet (Chen & Ong, 2022), CHGNet (Deng et al., 2023), MACE-MP-0 (Batatia et al., 2025), SevenNet (Park et al., 2024), and ORB (Neumann et al., 2024).

While these foundation MLIPs provide broad generalization, task-specific adaptation often requires a fine-tuning process. Generally, existing studies have employed full fine-tuning, sometimes with partial layer freezing to reduce computational cost (Gardner et al., 2025; Kong et al., 2025; Liu et al., 2026; Radova et al., 2025). Recently, equivariant adapters (Zhao et al., 2025) and ELoRA (Lu et al., 2025; Wang et al., 2025) have been proposed for equivariant architectures.

**Sparse Neural Networks and Sparsity-promoting Approaches.** Sparse neural networks, which maintain only a fraction of their weights, offer significant benefits such as model compression, inference acceleration, and improved generalization with reduced model complexity (Hoefler et al., 2021; Louizos et al., 2017; Molchanov et al., 2017; Roh et al., 2022; Wen et al., 2016). Early work by Han et al. (2015); Li et al. (2017); Srinivas & Babu (2015) proposed to prune trained networks and then fine-tune to recover the accuracy. Subsequently, the lottery ticket hypothesis (Frankle & Carbin, 2019) and Liu et al. (2019) shifted focus toward the subnetworks obtained after pruning and their initialization strategies (You et al., 2020; Zhang et al., 2021). Some studies (Azarian et al., 2020; LIU et al., 2020; Louizos et al., 2018; Savarese et al., 2020; XIAO et al., 2019) proposed to simultaneously learn weights and their sparse masks to avoid multiple rounds of model training and pruning. Soft threshold weight reparameterization (STR, Kusupati et al. (2020)), which we employ and describe below, belongs to this line of methods.

The concept of sparsity is particularly powerful in scientific ML. For example, the family of SINDy frameworks (Brunton et al., 2016) leverages sparsity to identify simple governing equations, based on the principle that physical phenomena are often governed by a few core variables (Kaheman et al., 2020; Mangan et al., 2019; 2016; Rudy et al., 2017). This reflects Occam's razor, where reducing model complexity enhances both interpretability and generalization in scientific applications. Despite these promising results, sparsity-promoting approaches remain largely underexplored in the MLIP domain (Sandberg et al., 2024; Torabi et al., 2026), particularly in fine-tuning scenarios.

## 3 METHOD

Our approach builds on equivariant GNNs (EGNNs), which are designed to preserve the inherent geometric symmetries in atomic systems. We first summarize their symmetry-preserving operations and then describe how we incorporate sparsity-promoting techniques into the fine-tuning of EGNNs.

### 3.1 PARAMETER UPDATES UNDER EQUIVARIANCE CONSTRAINTS

As MLIPs are designed to model atom-atom interactions, a key requirement is that their internal representations preserve the geometric symmetries of atomic systems, ensuring predictions remain consistent under changes of the coordinate system. This principle reflects the underlying spherical symmetry of atomic orbitals, which motivates the adoption of equivariant models such as EGNNs.

**Equivariant Representation.** In EGNNs, node and edge features are expressed as irreducible representations (irreps) of the rotation group, indexed by their order $\ell$: $\ell = 0$ for scalars, $\ell = 1$ for

vectors, and $\ell = 2$ for rank-2 tensors. When two input irreps are combined, the admissible output orders $\ell_{out}$ must satisfy $|\ell_{in1} - \ell_{in2}| \leq \ell_{out} \leq \ell_{in1} + \ell_{in2}$. These couplings, or *symmetry-allowed interaction paths* (Geiger & Smidt, 2022), are realized through Clebsch–Gordan coefficients (CGCs), which are predefined, non-trainable constants that mathematically guarantee equivariance.

The strength of each symmetry-allowed interaction path is modulated by a learnable scalar weight, and the collection of all such scalars forms a *path-specific* weight tensor $W$. Equivariance is strictly enforced by the predefined CGCs, while the capacity to learn lies entirely in the weights $W$. Therefore, when adapting a pre-trained model, the natural intervention point is this path-weight tensor. In other words, updating the path-weight tensor does not affect equivariance constraints. For a detailed explanation of the equivariant representation, we refer the reader to Appendix A.

**Weight Decomposition for Fine-tuning.** To adapt a foundation MLIP to a target domain, we decompose the path weights $W'$, to be fine-tuned, into frozen and trainable components:

$$W' = W + \Delta W, \tag{1}$$

where $W$ denotes the *frozen* weights from the pre-trained model and $\Delta W$ contains the adaptation parameters. From this perspective, fine-tuning corresponds to reweighting the relative contributions of interaction paths, thereby adjusting the model to the characteristics of the target dataset (*e.g.*, composition, pressure/temperature regime, or the level of theory used in DFT calculations).

Existing approaches such as ELoRA (Wang et al., 2025) parameterize $\Delta W$ as the product of two low-rank matrices, leading to dense updates in which every interaction path is perturbed to some degree. We hypothesize that more effective adaptation can be achieved through selective updates, enabling the model to adaptively identify and modify only the most critical paths.

**Sparse Adaptation in Practice.** To this end, we promote sparsity in $\Delta W$ by encouraging most path coefficients to remain close to zero, thereby suppressing unnecessary degrees of freedom during fine-tuning. Because $\Delta W$ modulates the learnable weights on CGCs in individual tensor interaction paths (see Appendix A), symmetry constraints are maintained, and equivariance is strictly preserved. In practice, we inject $\Delta W$ directly into the path-weight tensor, introducing negligible computational overhead (see Section 5.5) while enabling flexible domain adaptation. Here, we note that sparsity is not introduced to reduce training-time compute or memory usage, since the pre-trained weights $W$ remain dense; rather, it serves to induce selective and physically meaningful updates in $\Delta W$.

## 3.2 Controlling Sparsity during Adaptation

To enable selective adaptation, we learn $\Delta W$ in a sparse manner based on STR (Kusupati et al., 2020), a sparsity-promoting approach originally introduced in the computer vision domain. In STR, a dynamic pruning mechanism is governed by a learnable, per-layer scalar parameter $\tau$, which controls the pruning threshold $\delta = g(\tau)$, where $g$ is the sigmoid function. This threshold removes entries of $\Delta W$ with magnitudes below the threshold during training, making it sparse and yielding compact updates. In our case, only a subset of path weights corresponding to specific interaction paths is updated during adaptation, highlighting only the physically relevant interactions while suppressing symmetry-trivial paths.

Our training procedure starts by initializing the threshold parameter $\delta$ for each layer. Then, $\Delta W$ is initialized from a narrow Gaussian distribution $\mathcal{N}(0, \sigma^2 I)$. This initialization is necessary because if $\Delta W$ were set to zero, the threshold mechanism would cause vanishing gradients and prevent any learning. Before the forward pass at timestep $t$, parameters with magnitudes below $\delta$ in $\Delta W_t$ are pruned by applying a sparsify function $\mathcal{S}$ as:

$$\Delta W_t \leftarrow \mathcal{S}(\Delta W_t, \delta_t) := \text{sign}(\Delta W_t) \odot \text{ReLU}(|\Delta W_t| - \delta_t), \tag{2}$$

where $\odot$ denotes the element-wise Hadamard product.

In our work, we further decouple the updates of $\tau$ and $\Delta W$. First, $\Delta W$ is updated as:

$$\Delta W_{t+1} \leftarrow (1 - \eta_t \lambda_\Delta) \Delta W_t - \eta_t \nabla_{\Delta W_t} \mathcal{L}_{\text{total}} \odot \text{Mask}\{|\Delta W_t| > 0\}, \tag{3}$$

where $\eta_t$ is the learning rate, $\lambda_\Delta$ is the weight decay for $\Delta W$ and $\text{Mask}$ passes gradient flows only through the non-pruned elements. Simultaneously, $\tau$ is updated independently:

$$\tau_{t+1} \leftarrow (1 - \eta_{\tau,t} \lambda_\tau) \tau_t - \eta_{\tau,t} \nabla_{\tau_t} \mathcal{L}_{\text{total}}, \tag{4}$$

where it is governed by its own learning rate $\eta_{\tau,t}$ and a separate weight decay parameter $\lambda_\tau$, allowing for fine-grained control over the final sparsity. We empirically found that applying the naive STR causes instability during the fine-tuning process in equivariant MLIPs due to the coupled decay of $\tau$ and $\Delta W$. Decoupling these updates resolves this issue and provides stable, controllable fine-tuning.

## 4 EXPERIMENT

We demonstrate the effectiveness of our adaptation method on MACE and NequIP potentials (Batatia et al., 2022; 2025; Kovács et al., 2025; Tan et al., 2025) through comparison with two major baselines: standard full-parameter fine-tuning and ELoRA (Wang et al., 2025). This section presents the experimental setup, including datasets and hyperparameters. In addition, we report experiments that extend the task to predicting magnetic properties, beyond the standard force and energy predictions.

### 4.1 DATASETS

We use four benchmark datasets in our experiments. By default, each dataset is randomly split into training, validation, and test sets with an 8:1:1 ratio.

**Inorganic crystals.** For inorganic crystals, we use nine subsets from the LAM benchmark (Peng et al., 2026), including both single-element and compound systems. Specifically, we employ Cu (Zhang et al., 2020), Sn (Chen, 2023), Ti (Xu et al., 2024), V (Wang, 2022), W (Wang et al., 2022), H2O-PD (Zhang, 2021), SSE-PBE (Huang et al., 2021), Ag∪Au-PBE (Wang et al., 2021), and Al∪Mg∪Cu (Jiang et al., 2021).

**Revised MD17.** For molecules, we use the revised MD17 (rMD17) dataset (Christensen & von Lilienfeld, 2020), which provides reliable energies and forces derived from tighter quantum-mechanical calculations compared to the original MD17 dataset (Chmiela et al., 2017). The rMD17 dataset contains snapshots from long molecular dynamics trajectories of ten small organic molecules, with 100,000 structures each. Following the evaluation protocol of rMD17, we perform training and testing using 1,000 samples for training and a distinct set for validation.

**TM-O-Spin.** We construct the TM-O-Spin dataset via DFT calculations using `Quantum ESPRESSO` (QE, Giannozzi et al. (2009)). The dataset covers transition metal systems, specifically elemental Mn, Ni, and Fe, as well as transition metal oxides MnO and NiO. We generate ~1,000 structures across varying pressures, magnetic orders, and random displacements. Here, we employ the ACBN0 functional scheme through a patched version of QE (Agapito et al., 2015; Giannozzi et al., 2009; Lee & Son, 2020; Yang et al., 2021) to properly account for strong electronic correlations in transition metal oxides. Details of DFT computations are provided in Appendix C.

**MP-mag.** Further, we construct MP-mag dataset, a magnetic subset of the Materials Project trajectory (MPTrj) dataset (Deng et al., 2023), which is a large-scale collection of relaxation trajectories derived from the Materials Project database (Jain et al., 2013). The MPTrj dataset comprises approximately 1.6 million bulk crystal configurations covering 89 elements and has been widely used for training foundation MLIPs such as `MACE-MP-0b3`. From MPTrj, we select systems in which at least one atom carries a magnetic moment larger than $0.1\ \mu_B$, and the total number of atoms per structure is less than 100. This filtering resulted in a focused set of magnetic crystals suitable for benchmarking spin-related tasks.

### 4.2 BASE MODELS AND TRAINING DETAILS

In our main fine-tuning experiments, we use `MACE-MP-0b3` (Batatia et al., 2025) and `MACE-OFF23` (Kovács et al., 2025) as foundation models. These architectures are updated and extended variants of the original MACE model (Batatia et al., 2022), pre-trained on the MPTrj and SPICE datasets (Deng et al., 2023; Eastman et al., 2023), respectively, and tailored to crystal and molecular domains. These models are commonly trained to reproduce the PES by minimizing energy, force, and optionally stress prediction losses. We also conduct experiments using NequIP-OAM-L (Tan et al., 2025) to demonstrate that our method generalizes across different equivariant MLIPs. Experimental details and results are provided in Appendix D.

To extend the applicability of MLIPs beyond energy and force predictions, we conduct additional experiments on predicting magnetic properties. Specifically, a set of spin-aware layers, which receives the final node and edge embeddings as inputs, is added on top of the foundation model and predicts vector-valued non-collinear magnetic moments on each atom $\hat{\mu}_i$ and edge-wise energy correction $\epsilon_{ij}$ arising from spin-spin exchange interactions. The total energy is the sum of the energy predicted from the foundation model and the spin contributions. Consequently, the total loss $\mathcal{L}_{\text{total}}$ is a weighted sum of the four properties: $\mathcal{L}_{\text{total}} = \alpha_E \mathcal{L}_E + \alpha_F \mathcal{L}_F + \alpha_V \mathcal{L}_V + \alpha_\mu \mathcal{L}_\mu$, where $\mathcal{L}_\mu$ denotes magnetic moment loss, and the $\alpha$ terms are the corresponding weight coefficients. Each loss term $\mathcal{L}_E$, $\mathcal{L}_F$, $\mathcal{L}_V$, and $\mathcal{L}_\mu$ is computed using Huber loss (Huber, 1964). For detailed methods and background on both magnetic-aware energy and magnetic moment estimation, see Appendix B.

For the full fine-tuning and ELoRA baselines, we adopted the optimal hyperparameters from Wang et al. (2025) to ensure a fair comparison. For our method, initial learning rates were selected via grid search: $1 \times 10^{-2}$ for rMD17, $1 \times 10^{-3}$ for LAM, and $5 \times 10^{-3}$ for MP-mag. The sparsity threshold $\lambda_\tau$ was set to 0.01 and 0.3 for the low and high sparsity regimes, respectively. To initialize $\Delta W$, we used an empirical value of $\sigma = 0.01$ for the initial weight distribution, and set the threshold parameter $\delta = 0.001$ in Equation 2.

Across all experiments, we used a batch size of 64 and a weight decay of $1 \times 10^{-8}$, and employed the schedule-free AdamW optimizer (Defazio et al., 2024; Loshchilov & Hutter, 2019). All models were trained on a single GPU, with results averaged over three independent runs using different random seeds. For magnetic systems (TM-O-Spin and MP-mag), we set the coefficients for energy, force, stress, and magnetic moment losses to an equal ratio of $1 : 1 : 1 : 1$; otherwise, we follow the ratios of the MACE foundation models (Batatia et al., 2025; Kovács et al., 2025).

## 5 RESULTS AND ANALYSIS

In this section, we present experimental results demonstrating that our method is comparable to or superior to baselines such as full fine-tuning and ELoRA across all domains. Remarkably, these gains are achieved by modifying fewer than 3 % of the total parameters, with some tasks requiring updates to as few as 0.5 %. In addition, we present a preliminary interpretation of the model, which, to the best of our knowledge, represents a novel contribution to fine-tuning MLIPs.

### 5.1 ADAPTATION ON MOLECULAR AND CRYSTAL SYSTEMS

We first evaluate our method on the fundamental task of adapting foundation models for energy and force predictions. To this end, we adapt the `MACE-OFF23` model on the rMD17 dataset (Christensen & von Lilienfeld, 2020) for the molecular domain and the `MACE-MP-0b3` model on a subset of the LAM benchmark (Peng et al., 2026) for the inorganic crystal domain. To compare the enhancements achieved by each method, we also measure the zero-shot accuracy of the foundation models and the accuracy of models trained from scratch. We report the mean absolute error (MAE) for energy and force predictions, alongside two sparsity metrics for our method. Total sparsity measures the fraction of all model parameters that remain unchanged during fine-tuning. Layer-wise sparsity quantifies sparsity within the updated parameters $\Delta W$, defined as the proportion of entries in $\Delta W$ that remain exactly zero. Our method is presented in two configurations, detailed in Section 5.2: a standard setup ($\lambda_\tau = 0.01$) denoted as L that yielded the best overall predictive performance, and a high-sparsity setup ($\lambda_\tau = 0.3$) denoted as H.

**Results on Organic Molecules.** The left side of Table 1 presents our results on the rMD17 dataset. First, the results of zero-shot foundation models and models trained from scratch highlight the necessity of fine-tuning MLIPs. Among the fine-tuning approaches, our method demonstrates competitive with the baselines. In the standard setup (L), it surpasses both full fine-tuning and ELoRA on eight out of the ten molecules. Even more impressively, our method in a high-sparsity setup (H) also outperforms ELoRA across all molecules, showing the robustness of our sparse adaptation strategy on diverse organic systems.

**Results on Inorganic Crystals.** The results on inorganic crystal datasets shown on the right side of Table 1 further confirm the strength of our method. In the standard setup, our method achieves results comparable to or superior to both full fine-tuning and ELoRA. The strength of our approach is particularly evident for systems with large initial zero-shot errors, indicating a significant dis-

Table 1: **Evaluation on the rMD17 (left) and the LAM benchmark (right) datasets**. Energy MAE (E), force MAE (F), and total sparsity values (Sp.) are reported in meV/atom, meV/Å, and percent (%), respectively. We evaluate our method in both low-sparsity (L, $\lambda_\tau = 0.01$) and high-sparsity (H, $\lambda_\tau = 0.3$) regimes. Additional results including standard deviation and layer-wise sparsity values are provided in Appendix D.

| rMD17 | | Zero shot | Scratch | Full | ELoRA | Ours (L) | Ours (H) | LAM | | Zero shot | Scratch | Full | ELoRA | Ours (L) | Ours (H) |
|---|---|---|---|---|---|---|---|---|---|---|---|---|---|---|---|
| Aspirin | E | 4.72 | 0.60 | 0.19 | 0.21 | 0.17 | 0.20 | Al∪Mg∪Cu | E | 238.42 | 27.90 | 1.33 | 1.63 | 1.83 | 2.23 |
| | F | 354.42 | 25.55 | 8.09 | 8.52 | 7.56 | 8.22 | | F | 43.17 | 8.23 | 5.59 | 6.82 | 6.10 | 6.63 |
| | Sp. | - | - | - | - | 83.19 | 96.84 | | Sp. | - | - | - | - | 97.61 | 99.62 |
| Benzene | E | 4.50 | 0.16 | 0.02 | 0.02 | 0.01 | 0.01 | Cu | E | 359.05 | 3.31 | 0.66 | 0.79 | 0.65 | 0.71 |
| | F | 233.45 | 7.63 | 1.67 | 0.94 | 0.90 | 0.96 | | F | 50.94 | 7.08 | 3.96 | 5.29 | 4.32 | 4.80 |
| | Sp. | - | - | - | - | 83.50 | 96.85 | | Sp. | - | - | - | - | 97.57 | 99.58 |
| Malonaldehyde | E | 7.47 | 0.98 | 0.20 | 0.24 | 0.21 | 0.22 | Sn | E | 31905.00 | 8.60 | 32.74 | 9.33 | 2.18 | 2.49 |
| | F | 405.37 | 22.66 | 7.91 | 8.61 | 7.58 | 8.03 | | F | 114.22 | 29.58 | 25.82 | 32.46 | 23.92 | 25.39 |
| | Sp. | - | - | - | - | 83.54 | 97.39 | | Sp. | - | - | - | - | 97.46 | 99.42 |
| Paracetamol | E | 5.33 | 0.44 | 0.12 | 0.13 | 0.11 | 0.13 | SSE-PBE | E | 20.33 | 0.53 | 0.29 | 0.41 | 0.33 | 0.32 |
| | F | 232.60 | 21.39 | 6.74 | 6.65 | 5.92 | 6.53 | | F | 82.38 | 12.74 | 8.13 | 10.85 | 8.92 | 9.78 |
| | Sp. | - | - | - | - | 83.29 | 96.92 | | Sp. | - | - | - | - | 96.77 | 99.33 |
| Toluene | E | 5.54 | 0.19 | 0.05 | 0.06 | 0.05 | 0.05 | Ti | E | 126.38 | 8.61 | 2.57 | 3.50 | 2.59 | 3.28 |
| | F | 280.54 | 11.78 | 3.31 | 3.01 | 2.70 | 2.92 | | F | 133.56 | 51.38 | 39.24 | 43.65 | 39.01 | 41.03 |
| | Sp. | - | - | - | - | 83.31 | 96.87 | | Sp. | - | - | - | - | 97.23 | 99.45 |
| Azobenzene | E | 4.96 | 0.28 | 0.10 | 0.11 | 0.08 | 0.10 | V | E | 226.44 | 12.09 | 1.45 | 2.65 | 1.67 | 3.16 |
| | F | 353.55 | 17.05 | 6.49 | 6.59 | 6.26 | 6.90 | | F | 168.02 | 41.26 | 23.15 | 31.25 | 25.38 | 29.26 |
| | Sp. | - | - | - | - | 83.20 | 97.30 | | Sp. | - | - | - | - | 97.60 | 99.65 |
| Ethanol | E | 6.93 | 0.33 | 0.08 | 0.10 | 0.09 | 0.10 | W | E | 378.04 | 7.22 | 2.94 | 4.33 | 3.18 | 4.03 |
| | F | 369.69 | 13.28 | 4.24 | 4.58 | 4.12 | 4.49 | | F | 449.23 | 96.13 | 66.00 | 89.26 | 71.71 | 79.01 |
| | Sp. | - | - | - | - | 83.21 | 96.32 | | Sp. | - | - | - | - | 97.44 | 99.45 |
| Naphthalene | E | 5.63 | 0.19 | 0.06 | 0.07 | 0.05 | 0.06 | Ag∪Au-PBE | E | 80.15 | 14.27 | 3.64 | 1.89 | 1.26 | 1.40 |
| | F | 288.43 | 13.59 | 3.75 | 3.34 | 3.25 | 3.26 | | F | 93.36 | 17.55 | 14.03 | 10.39 | 8.55 | 9.24 |
| | Sp. | - | - | - | - | 83.29 | 97.33 | | Sp. | - | - | - | - | 97.38 | 99.41 |
| Salicylic | E | 6.13 | 0.32 | 0.09 | 0.08 | 0.08 | 0.08 | H$_2$O-PD | E | 311.19 | 5.10 | 11.67 | 4.98 | 3.67 | 4.84 |
| | F | 351.31 | 20.87 | 5.99 | 5.87 | 5.51 | 5.72 | | F | 346.92 | 80.64 | 84.25 | 69.41 | 62.31 | 67.71 |
| | Sp. | - | - | - | - | 83.05 | 97.17 | | Sp. | - | - | - | - | 97.47 | 99.48 |
| Uracil | E | 6.91 | 0.41 | 0.08 | 0.08 | 0.07 | 0.08 | | | | | | | | |
| | F | 365.86 | 17.41 | 6.03 | 4.96 | 4.52 | 4.92 | | | | | | | | |
| | Sp. | - | - | - | - | 83.69 | 97.44 | | | | | | | | |

tribution shift from the pre-training data (*e.g.*, Sn and H$_2$O-PD). This was achieved by modifying approximately 3 % of the model's parameters. For the high-sparsity setup, by updating merely 0.5-0.7 % of the total parameters, our method still demonstrates comparable accuracy to full fine-tuning and better prediction accuracy than ELoRA for energy and forces across nearly all subsets. This highlights our method's ability to achieve performance gains through highly targeted and compact modifications. These results suggest that our approach is not confined to a single material class but can generalize across diverse chemical domains, from inorganic crystals to organic molecules. To further verify that our method is not specific to the MACE architecture, we additionally evaluate on the NequIP-OAM-L architecture (Tan et al., 2025). As shown in Table I in Appendix D, our method achieves comparable or superior accuracy to full fine-tuning and ELoRA across all benchmark datasets, demonstrating its generalizability across different equivariant architectures.

## 5.2 Ablation Studies

Our framework allows selective adaptation in two directions: by targeting specific model layers and by controlling the pruning strength via the threshold weight decay $\lambda_\tau$. We conduct ablation studies to explore both aspects and describe our experimental design, then report key findings.

First, we leverage our framework's ability to perform module-specific tuning. We investigate three distinct strategies: (1) updating only the linear layers, (2) only the fully-connected tensor product (FCTP) layers, or (3) both, which we refer to as *Linear*, *FCTP*, and *All* settings.

As shown in Figure 1, the predictive performance trends for the *Linear* and *All* settings are nearly identical across a wide range of $\lambda_\tau$ values. Conversely, the *FCTP* setting consistently yields the poorest performance, indicating that the majority of the adaptation capability is captured by linear layers. Results on TM-O-Spin (shown in Figure II in Appendix D) demonstrate a similar tendency.

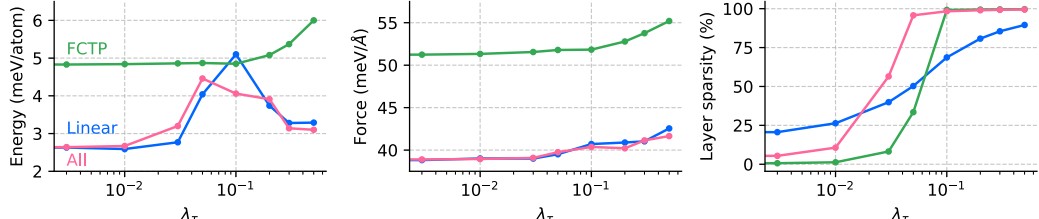

Figure 1: **Ablation study on the effect of the threshold weight decay ($\lambda_\tau$) for Ti dataset.** Energy and force MAEs along with layer-wise sparsity are plotted against varying $\lambda_\tau$ values. Three different update strategies are compared: updating only the linear layers, only the FCTP layers, or both layers (denoted as *All*). The tendency of the linear and *All* configurations is similar and significantly superior to that of *FCTP*, showing a stable, high-sparsity regime at large $\lambda_\tau$ values.

Table 2: **Evaluation on TM-O-Spin dataset.** Energy (meV/atom), force (meV/Å), and magnetic moment ($\mu$) MAEs along with sparsity values (%) are presented for comparison with the baselines.

| Metric | Scratch | Full | Method ELoRA | Ours (L) | Ours (H) |
|---|---|---|---|---|---|
| Energy | $27.28_{\pm 6.73}$ | $11.58_{\pm 2.87}$ | $12.44_{\pm 1.88}$ | $9.50_{\pm 0.11}$ | $10.57_{\pm 1.18}$ |
| Force | $174.70_{\pm 17.68}$ | $96.89_{\pm 47.09}$ | $116.31_{\pm 2.70}$ | $70.46_{\pm 28.83}$ | $74.15_{\pm 33.76}$ |
| Magnetic moment | $0.035_{\pm 0.013}$ | $0.029_{\pm 0.008}$ | $0.038_{\pm 0.014}$ | $0.028_{\pm 0.007}$ | $0.030_{\pm 0.006}$ |
| Sparsity (total) | - | - | - | $90.21_{\pm 0.01}$ | $91.83_{\pm 0.01}$ |
| Sparsity (layer) | - | - | - | $43.17_{\pm 0.40}$ | $89.70_{\pm 0.29}$ |

However, a closer look at the module sparsity trends reveals a key difference between the *Linear* and *All* settings. As shown in Figure 1, for the Linear-only strategy, sparsity increases gradually with increasing $\lambda_\tau$. In contrast, for the *All* strategy, module sparsity increases sharply at a much earlier stage. This suggests that the linear parameters are more critical for maintaining performance and are thus pruned more reluctantly by the model, a point we revisit in our interpretation section.

Furthermore, the energy plots for both the *Linear* and *All* settings commonly exhibit two distinct, effective adaptation regimes. First, a low-sparsity or high-performance regime (labeled as Ours (L) in Table 1) exists at small $\lambda_\tau$ with low module sparsity. For our main experiments, we selected the optimal point from this regime, defined as the $\lambda_\tau$ value yielding the best performance with the most compact update. This corresponds to $\lambda_\tau = 0.01$ for most datasets, except for the TM-O-Spin, where the optimum occurs at $\lambda_\tau = 0.05$.

Second, we identified a high-sparsity, stable regime (labeled as Ours (H)) at larger $\lambda_\tau$ from 0.3, which maintains strong performance with extreme sparsity despite being slightly sub-optimal. To represent this regime, we consistently choose $\lambda_\tau = 0.3$ across all datasets. We hypothesize that this high-sparsity regime forces the model to distill the adaptation into the most essential parameter updates. This provides hints for interpretability, as we will explore further in Section 5.4.

### 5.3 EXTENSION TOWARDS MAGNETIC SYSTEMS

Magnetism often emerges in transition-metal systems, where spin degrees of freedom play a central role. The relevant energy scale is typically on the order of 10 meV/atom, making accurate prediction essential. Extending non-magnetic foundation models is therefore a highly efficient strategy, as it enables the reuse of pre-trained knowledge while capturing the subtle energy differences associated with magnetic configurations. We add spin-aware layers (+8.6 % parameters) to MACE-MP-0b3, which are trained from scratch, while our sparsity-promoting adaptation is applied exclusively to the original foundation model parameters. Further architectural details are provided in Appendix B.

We first evaluate this approach using our custom TM-O-Spin dataset. As shown in Table 2, fine-tuning from the pre-trained model provides substantial improvements over training from scratch, underscoring the importance of domain-specific fine-tuning. While predictions for spin-aware energy showed only slight improvements, the accuracy of total energy and forces improved significantly,

Table 3: **Evaluation on the MP-mag dataset**. Energy (meV/atom), force (meV/Å), and magnetic moment ($\mu$) MAEs along with sparsity values (%) are presented for comparison with the baselines.

| Method | Metric | | | | |
| --- | --- | --- | --- | --- | --- |
| | Energy | Force | Magnetic moment | Sparsity (total) | Sparsity (layer) |
| Full | $18.75_{\pm0.80}$ | $37.56_{\pm0.16}$ | $0.015_{\pm0.000}$ | - | - |
| ELoRA | $17.37_{\pm0.81}$ | $40.34_{\pm0.21}$ | $0.013_{\pm0.000}$ | - | - |
| Ours | | | | | |
| └ Linear, $\lambda_\tau = 0.01$ | $17.56_{\pm0.89}$ | $42.11_{\pm0.35}$ | $0.012_{\pm0.000}$ | $90.15_{\pm0.04}$ | $41.27_{\pm1.08}$ |
| └ Linear, $\lambda_\tau = 0.3$ | $18.40_{\pm0.86}$ | $39.60_{\pm0.37}$ | $0.014_{\pm0.000}$ | $91.83_{\pm0.01}$ | $89.79_{\pm0.36}$ |
| └ All, $\lambda_\tau = 0.05$ | $16.56_{\pm0.71}$ | $40.75_{\pm0.71}$ | $0.011_{\pm0.000}$ | $30.78_{\pm0.12}$ | $20.90_{\pm0.15}$ |
| └ All, $\lambda_\tau = 0.3$ | $17.20_{\pm0.75}$ | $39.59_{\pm0.45}$ | $0.013_{\pm0.000}$ | $87.53_{\pm0.04}$ | $94.00_{\pm0.05}$ |

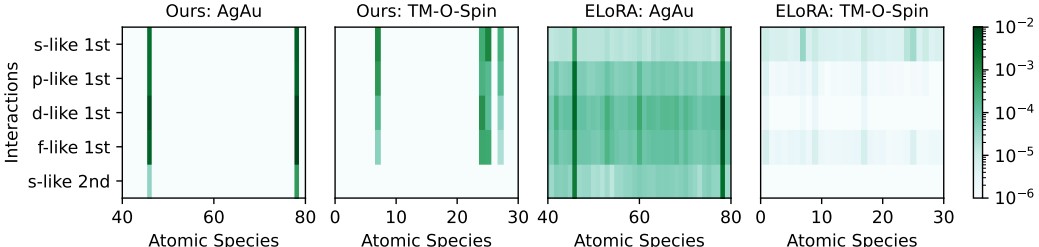

Figure 2: **Comparison of parameter update patterns of our method and ELoRA, fine-tuned on the TM–O–Spin dataset**. The y-axis denotes interaction channels corresponding to orbital-like symmetries ($s$, $p$, $d$, $f$) at different layers: *1st* indicates the first layer and *2nd* the second layer. The $x$-axis shows atomic species indices, and colors represent the magnitude of parameter changes quantified by $1 - R^2$ (coefficient of determination), with darker colors indicating stronger updates.

indicating the effectiveness of our approach. Moreover, compared to the baselines, our method with the high-sparsity setting (*i.e.*, Ours (H)) attains accuracy comparable to full fine-tuning while substantially outperforming ELoRA. An ablation study shows trends consistent with those observed on the Ti dataset (see Figure II in Appendix D).

Furthermore, to assess scalability, we evaluate our approach on the large-scale MP-mag dataset, shown in Table 3. On this broader and more diverse dataset, differences in predictive performance among training methods diminished. Full fine-tuning achieved the best force prediction accuracy, while our *All* setting with low sparsity ($\lambda_\tau = 0.05$) was optimal for other metrics. This suggests that dense updates become more competitive when trained on sufficiently large and diverse in-domain datasets, revealing the practical trade-offs among different fine-tuning strategies.

## 5.4 MODEL INTERPRETATION

Our sparsity-promoting framework provides a fundamental interpretation of the fine-tuned model, enabling us to identify which physical interaction pathways are most critical for adapting to a new task. We analyze the updated parameters from the high-sparsity *All* setting, as it results in highly selective changes across both linear and FCTP layers. We quantify the magnitude of these changes, especially for FCTP layers, by measuring the $R^2$ difference of the path-wise weights before and after fine-tuning.

Primary results from TM–O–Spin dataset, shown in Figure 2, reveal physically intuitive adaptation patterns. When fine-tuned with our method, the model exhibits prominent parameter updates in pathways associated with $p$- and $d$-orbital-like features for the transition metals, and $s$- and $p$-like features for oxygen. This aligns with fundamental physical principles that the valence electrons of transition metals occupy $d$-orbitals, while those of oxygen are in the $s$- and $p$-orbitals, which govern their chemical bonding. In contrast, ELoRA produces parameter updates for elements outside the training set and modifies all layers, which may not be necessary. Similar update patterns are also observed in the AgAu dataset.

Such an update pattern arises from the low-rank construction of $\Delta W$ in ELoRA, where $\Delta W = AB$. In this formulation, any parameter update applied to $A$ or $B$ propagates across entire rows or columns of $\Delta W$. Since these low-rank factors are shared across all atomic species channels, updates originating from species present in the training data inevitably propagate to parameters for all atomic species in the reconstructed $\Delta W$, leading to a diffuse, non-specific update pattern that hinders clear interpretation. This demonstrates that our sparsity-promoting approach can provide fundamental interpretability, highlighting how a foundation model calibrates its physical knowledge to new domains.

## 5.5 COMPUTATIONAL COST

We measured the average training iteration time and peak memory consumption for full fine-tuning, ELoRA, and our method, as shown in Table 4. The experiments were conducted on the rMD17 Aspirin dataset using the `MACE-OFF23` model with a single NVIDIA H100 GPU. We note that during inference, $\Delta W$ is merged with $W$, resulting in identical execution speed and memory footprint to those of the

Table 4: **Computational cost of each method.** We report the average training iteration time (msec) and peak memory consumption (GB) of each fine-tuning method.

|  | Full | ELoRA ($r = 16$) | Ours (Linear) | Ours (FCTP) | Ours (All) |
|---|---|---|---|---|---|
| Time (msec) | 144.92 | 165.56 | 159.49 | 149.77 | 161.75 |
| Memory (GB) | 5.17 | 5.67 | 5.18 | 5.69 | 5.70 |

original model; therefore, only training iteration time and memory consumption are measured. In Appendix D, we further provide a comparison of convergence speed across the three methods.

**Training Iteration Time.** Our method exhibits a small overhead in training iteration time, typically 3-11 % slower than full fine-tuning. However, it outperforms ELoRA in terms of training speed, as the matrix multiplications required to reconstruct $\Delta W$ in ELoRA create a computational bottleneck.

**Memory Usage.** In the case of our *Linear* setting, the memory overhead is marginal compared to full fine-tuning. While the *FCTP* and *All* settings exhibit increased memory usage relative to full fine-tuning, they remain comparable to that of ELoRA. This behavior stems from the distinctive training mechanism of MLIPs. In conventional neural network training, memory consumption is often dominated by optimizer states that scale with the number of parameters, where ELoRA achieves memory savings by reducing this overhead. In contrast, MLIP training is primarily dominated by intermediate activations and gradients required for *autograd*-based force prediction. Consequently, the size of the optimizer state has only a minor influence on the overall memory footprint, resulting in comparable memory usage between ELoRA and our method.

## 6 CONCLUSION AND DISCUSSION

In this paper, we propose a novel sparsity-promoting fine-tuning method tailored for equivariant MLIPs. By updating a small subset of parameters, as few as 0.5 % in some cases, our method achieves comparable or superior accuracy to full parameter fine-tuning and existing equivariant fine-tuning approach. Furthermore, we demonstrate its versatility by extending a foundation MLIP to the challenging task of magnetic-aware energy and magnetic moment prediction. These results present the promise of extending foundation models to broader tasks beyond traditional force and energy prediction. In addition, our sparsity-promoting approach offers interpretability, clearly revealing physically meaningful insights into the fine-tuned model. Taken together, our approach suggests a promising direction for interpretable adaptation of materials foundation models.

**Outlook.** While fine-tuning is a common paradigm, its inherent nature often limits computational efficiency gains, one of the key advantages associated with sparse neural networks. In light of this, structured sparse pre-training represents a promising direction toward providing MLIPs with both computational efficiency and enhanced interpretability (Wen et al., 2016). Such an approach would enable genuine hardware acceleration through structured sparsity patterns, allowing MLIPs to perform large-scale simulations faster than current dense implementations. Furthermore, sparse pre-training presents opportunities for designing sparsity-aware interpretable MLIP architectures fundamentally distinct from current MLIPs, building upon the physically meaningful basis selection demonstrated in our work. These directions suggest that the intersection of structured sparsity and interpretable design could define the next generation of equivariant MLIPs.

## Reproducibility Statement

To enhance reproducibility of experiments in this work, we described the overall experimental setup in Section 4. For the magnetism-related experiments, we constructed a dataset through our own DFT calculations and extended the MACE model architecture with spin-prediction layers to predict magnetic properties, with the details provided in Appendix B and C. The implementation code for our proposed method is included in the Supplementary Materials.

## Ethics Statement

This work presents a sparsity-promoting fine-tuning method for equivariant MLIPs and does not raise specific ethical concerns. Our research does not involve human subjects, sensitive data, or potential dual-use applications.

## Acknowledgments

This research was supported by the National Research Foundation of Korea (NRF) grants (RS-2021-NR05515, RS-2024-00336576, RS-2023-0022663, RS-2024-00342044, RS-2025-16063688, RS-2025-00555621, RS-2025-02293161), the Institute of Information & Communications Technology Planning & Evaluation (IITP) grants (RS-2019-II190075, RS-2022-II220264, RS-2024-00353131, RS-2025-02653113), the National Supercomputing Center (KSC-2025-CRE-0176), the High-Performance Computing Support Project funded by the Korea government. This work was also supported by Samsung Electronics, the SOFT Foundry Institute at SNU, the Youlchon Foundation, the Ewha Global Excellence Program Research Grant, and the Korea Institute for Advanced Study, including computing resources from the Center for Advanced Computation in KIAS.

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

## A  BACKGROUND FOR EQUIVARIANT OPERATIONS IN EGNNS

**Tensor Products in EGNNs and Adaptation Strategy.** The fundamental operations in EGNNs, such as message passing and feature updates, are performed using tensor product operations in Equation 5. These operations are typically implemented using `e3nn` library (Geiger & Smidt, 2022), and are designed to guarantee two crucial symmetries required for atomistic systems: **permutation invariance** and $E(3)$-**equivariance**. Permutation invariance is automatically satisfied by the inherent nature of message passing on graphs, as message aggregation is invariant under node permutation (see Satorras et al. (2021) for more details).

To satisfy $E(3)$-equivariance (invariance to translation, and equivariance to rotation and reflection), EGNNs for atomic systems typically use relative position vectors between atoms as input, which are inherently translationally invariant. Rotational equivariance is then handled by representing features as geometric objects called *irreducible representations*, which are tensors of a specific *order* or *type-$\ell$*. This concept is physically analogous to atomic orbitals:

- An $\ell = 0$ feature is a scalar (like an $s$-orbital), remaining invariant under rotation.
- An $\ell = 1$ feature is a vector (like a $p$-orbital), which transforms linearly under rotations.
- An $\ell = 2$ feature is a rank-2 tensor (like a $d$-orbital), transforming according to higher-order rotation rules.

Furthermore, each feature is indexed with the magnetic quantum number $m$, with $-\ell \leq m \leq \ell$, and a channel index $k$, which is the same concept of channels in deep learning, allowing the model to learn multiple distinct features for each geometric type.

For a general formulation of interactions, we start with a tensor product (TP) operation between two tensors $\boldsymbol{u}$ and $\boldsymbol{v}$ with orders $\ell_1$ and $\ell_2$. Specific TPs are defined per interaction path $(\ell_1, \ell_2 \rightarrow \ell_3)$, which generates an output feature of type $\ell_3$. This is physically analogous to the orbital interactions, for example, interaction between $p$-orbital like features $(\ell_1 = \ell_2 = 1)$ can contribute to $s$, $p$, and $d$-orbital-like features, as dictated by the triangle inequality: $|\ell_1 - \ell_2| \leq \ell_3 \leq \ell_1 + \ell_2$. The general form of the tensor product for a given interaction path is:

$$\phi_{k_3,\ell_3,m_3}^{(\ell_1,\ell_2 \rightarrow \ell_3)}(\boldsymbol{u} \otimes \boldsymbol{v}) = \sum_{k_1,k_2} \sum_{m_1,m_2} W_{\ell_3\ell_2\ell_1}^{k_3k_2k_1} C_{\ell_1 m_1, \ell_2 m_2}^{\ell_3 m_3} u_{k_1\ell_1 m_1} v_{k_2\ell_2 m_2}, \quad (5)$$

where $\phi$ denotes a TP layer in EGNNs, $C_{\ell_1 m_1, \ell_2 m_2}^{\ell_3 m_3}$ are the predefined CGCs, which enforce equivariance of individual interaction paths, derived from group theory. Finally, the learnable weight tensor $W_{\ell_3\ell_2\ell_1} \in \mathbb{R}^{k_3 \times (k_1 \times k_2)}$ models how strongly each interaction path should contribute to the final output, based on the training data. For further information, see Satorras et al. (2021), Batatia et al. (2022), and Geiger & Smidt (2022).

**Adaptation of Equivariant MLIPs.** As established in Equation 5, the equivariance of the network is mathematically guaranteed by the CGCs, while the learnable weights $W$ simply modulate the strength of each symmetry-allowed interaction path. Based on this principle, we can adapt the model to a new domain by modifying path-wise weights $W_{\ell_3\ell_2\ell_1}^{k_3k_2k_1}$ without breaking the model's fundamental equivariance.

## B  DETAILS IN SPIN LAYERS

We extend the baseline MACE architecture with a minimal set of spin-aware layers, denoted as the `SpinProductBlock` ($f_\theta$), which processes atomic features from the foundation model, alongside spin information. These new layers take spin orientations as an additional input to predict atomic magnetic moments and their energetic contributions. If spin inputs are not provided, these layers are deactivated, allowing the model to operate as the standard non-magnetic MLIP.

In this design, the model receives the spin *directions* as input (encoded as unit vectors) and the model predicts the magnitude of the magnetic moment during readout. The raw spin directional vectors are first embedded into an equivariant representation through a linear map from the physical spin irrep ($1o$) to a higher-dimensional space with hidden irreps of $16 \times (0e + 1o + 2e + 3o)$. These embedded spin features are combined with the node and edge hidden features $\mathbf{x}_i$ and $\mathbf{e}_{ij}$ from the

MACE backbone via a fully connected tensor product, producing *spin-augmented node and edge features* $\mathbf{x}'_i$ and $\mathbf{e}'_{ij}$ that incorporate both local atomic environments and spin degrees of freedom.

From these spin-augmented features, the spin layers predict per-atom spin vectors $\mathbf{S}_i$ and edge-level energy contributions $\epsilon_{ij}$ as

$$(\mathbf{S}_i, \epsilon_{ij}) = f_\theta\left(\mathbf{x}'_i \, ; \, \mathbf{e}'_{ij}\right), \qquad (6)$$

with $i, j \in \{1, \ldots, N\}$ in a system of $N$ atoms. The edge-level contributions are subsequently aggregated across the graph,

$$E_{\text{spin}} = \sum_{(i,j) \in \mathcal{E}} \epsilon_{ij}, \qquad (7)$$

yielding the spin-induced energy contribution $E_{\text{spin}}$. The total energy can then be expressed as

$$E_{\text{total}} = E_0 + E_{\text{spin}}, \qquad (8)$$

where $E_0$ denotes the energy predicted by the foundation model. Finally, the framework predicts the magnetic moment $\mathbf{S}_i$ associated with each atom.

This formulation can be interpreted as learning *effective exchange-like interactions* directly from data: the function $f_\theta$ acts as a flexible mapping that generalizes beyond fixed analytic forms while remaining sensitive to spin-dependent features. Importantly, as spin vectors enter only through their equivariant embeddings, the model automatically preserves *time-reversal symmetry (TRS)*: flipping all spins $\mathbf{S}_i \mapsto -\mathbf{S}_i$ leaves the energy invariant. In this sense, adding spin layers provides a data-driven representation of spin interactions that captures complex spin-lattice couplings while ensuring essential physical symmetries.

## C  DFT CALCULATIONS FOR TM-O-SPIN DATASET CONSTRUCTION

The TM-O-Spin dataset was constructed via DFT calculations using the QE (Giannozzi et al., 2009) package. Norm-conserving pseudopotentials from the `PseudoDojo` library (van Setten et al., 2018) within the PBE exchange-correlation functional were employed. A plane-wave cutoff of 100 Ry was used, and Brillouin zone sampling was performed with a uniform $k$-point mesh generated using a spacing of $0.05$ Å$^{-1}$. All calculations were carried out in the collinear-spin framework with a self-consistent convergence threshold of $10^{-8}$ Ry.

To properly account for strong electronic correlations Hubbard $U$ in transition-metal oxides, we employed the *ACBN0 functional* scheme, which *automatically and self-consistently determines the on-site Hubbard $U$* from the electronic structure (Agapito et al., 2015; Lee & Son, 2020; Yang et al., 2021). The Hubbard $U$ parameters were evaluated in the G-type antiferromagnetic ground state of the corresponding transition-metal oxide (MnO and NiO). This was realized through a patched version of QE (Giannozzi et al., 2009; Lee & Son, 2020; Yang et al., 2021).

## D  ADDITIONAL RESULTS

This section provides additional experiments and analyses. We first evaluate our method on the NequIP-OAM-L architecture (Tan et al., 2025) to demonstrate generalizability across different equivariant architectures (Table I), and then analyze the convergence behavior of our method compared to baselines (Figure I). We also include an ablation study on the threshold weight decay ($\lambda_\tau$) for the TM-O-Spin dataset (Figure II) and detailed evaluation results on the rMD17 (Table II) and LAM (Table III) benchmark datasets.

**Results on NequIP Foundation Model.** To demonstrate that our method generalizes across different equivariant architectures, we additionally evaluated on the NequIP-OAM-L model (Batzner et al., 2022; Tan et al., 2025), another *state-of-the-art* foundation MLIP built on a different architecture from MACE. Following the main experiments on MACE models, we evaluated on the same rMD17 and LAM benchmark datasets. Here, we adjusted the learning rates for the NequIP-OAM-L architecture, using $5 \times 10^{-3}$ for full fine-tuning and $1 \times 10^{-2}$ for our method. The results presented in Table I indicate that our method achieves performance comparable to or surpassing that of the full fine-tuning baseline.

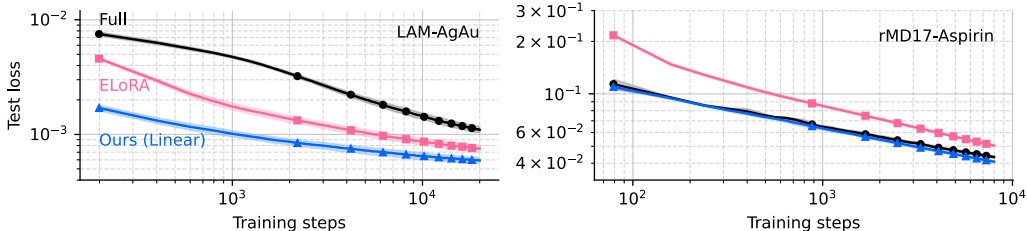

Figure I: **Convergence speed analysis comparing full fine-tuning, ELoRA, and our method.** We plot the total loss on the test data against the training steps for the Ag∪Au-PBE and Aspirin dataset. The lines represent averages over three independent experiments, and the shaded regions indicate the corresponding standard deviations.

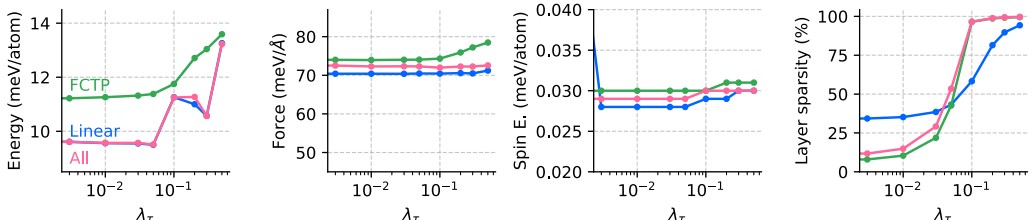

Figure II: **Ablation study on the effect of the threshold weight decay ($\lambda_\tau$) for TM-O-Spin dataset.** Energy MAE, force MAE, and layer-wise sparsity are plotted against varying $\lambda_\tau$ values.

**Analysis of Convergence Speed.** We analyzed the convergence behavior of our method (*Linear* setting) in comparison to full fine-tuning and ELoRA. For this evaluation, we selected the rMD17 Aspirin dataset and the Ag∪Au-PBE dataset as representative examples of organic and inorganic systems. The baseline configurations are consistent with those used in the main experiment, and all models were trained for 500 epochs with their respective optimal hyperparameters. As shown in Figure I, our method achieves significantly faster convergence on the Ag∪Au-PBE dataset compared to both full fine-tuning and ELoRA. On the Aspirin dataset, our method exhibits a modest but consistent advantage in convergence speed over the baselines.

## E   USE OF LARGE LANGUAGE MODEL

Large language models were used solely for minor English grammar corrections. They were not involved in the research ideation, experimental design, analysis, or substantive writing.

Table I: **Evaluation on the rMD17 and LAM benchmark datasets, with NequIP-OAM-L as the foundation model.** Energy MAE (E) and force MAE (F) values with standard deviation are reported in meV/Å and meV/atom, respectively. Sparsity (Sp.) values are reported as total sparsity with layer-wise sparsity in parentheses, in percent (%). We evaluate our method with the sparsity of ($\lambda_\tau = 0.03$) for rMD17 dataset, and ($\lambda_\tau = 0.003$) for LAM datasets.

| rMD17 | | Full | Method ELoRA | Ours | LAM | | Full | Method ELoRA | Ours |
|---|---|---|---|---|---|---|---|---|---|
| Aspirin | E | $0.37_{\pm0.01}$ | $0.38_{\pm0.05}$ | $0.33_{\pm0.04}$ | Al∪Mg∪Cu | E | $1.33_{\pm0.16}$ | $1.17_{\pm0.17}$ | $1.10_{\pm0.15}$ |
| | F | $16.54_{\pm0.88}$ | $20.50_{\pm0.21}$ | $16.54_{\pm0.27}$ | | F | $5.19_{\pm0.37}$ | $5.22_{\pm0.34}$ | $5.29_{\pm0.35}$ |
| | Sp. | - | - | $98.22_{(71.87)}$ | | Sp. | - | - | $93.90_{(3.73)}$ |
| Benzene | E | $0.16_{\pm0.06}$ | $0.05_{\pm0.02}$ | $0.05_{\pm0.00}$ | Cu | E | $0.67_{\pm0.06}$ | $0.35_{\pm0.01}$ | $0.49_{\pm0.05}$ |
| | F | $9.43_{\pm0.28}$ | $4.59_{\pm0.58}$ | $4.26_{\pm0.34}$ | | F | $2.81_{\pm0.05}$ | $2.76_{\pm0.02}$ | $3.11_{\pm0.06}$ |
| | Sp. | - | - | $98.80_{(81.04)}$ | | Sp. | - | - | $94.23_{(8.87)}$ |
| Malonaldehyde | E | $0.35_{\pm0.01}$ | $0.42_{\pm0.10}$ | $0.35_{\pm0.01}$ | Sn | E | $1.55_{\pm0.03}$ | $1.60_{\pm0.06}$ | $1.56_{\pm0.04}$ |
| | F | $13.52_{\pm0.82}$ | $14.64_{\pm0.42}$ | $13.60_{\pm0.30}$ | | F | $20.83_{\pm0.52}$ | $20.94_{\pm0.45}$ | $20.08_{\pm0.46}$ |
| | Sp. | - | - | $98.40_{(74.75)}$ | | Sp. | - | - | $93.86_{(3.14)}$ |
| Paracetamol | E | $0.29_{\pm0.00}$ | $0.40_{\pm0.06}$ | $0.25_{\pm0.04}$ | SSE-PBE | E | $0.21_{\pm0.01}$ | $0.22_{\pm0.02}$ | $0.24_{\pm0.02}$ |
| | F | $15.13_{\pm0.38}$ | $21.59_{\pm0.87}$ | $15.82_{\pm0.91}$ | | F | $6.40_{\pm0.21}$ | $6.91_{\pm0.26}$ | $7.58_{\pm0.20}$ |
| | Sp. | - | - | $98.26_{(72.47)}$ | | Sp. | - | - | $94.41_{(11.81)}$ |
| Toluene | E | $0.18_{\pm0.01}$ | $0.16_{\pm0.03}$ | $0.14_{\pm0.02}$ | Ti | E | $1.77_{\pm0.06}$ | $1.62_{\pm0.12}$ | $1.58_{\pm0.05}$ |
| | F | $11.85_{\pm0.40}$ | $8.35_{\pm0.23}$ | $8.92_{\pm1.02}$ | | F | $33.02_{\pm1.08}$ | $33.02_{\pm2.33}$ | $30.82_{\pm1.27}$ |
| | Sp. | - | - | $98.52_{(76.59)}$ | | Sp. | - | - | $93.89_{(3.47)}$ |
| Azobenzene | E | $0.24_{\pm0.00}$ | $0.27_{\pm0.01}$ | $0.17_{\pm0.01}$ | V | E | $1.16_{\pm0.08}$ | $0.88_{\pm0.07}$ | $0.78_{\pm0.05}$ |
| | F | $15.31_{\pm0.36}$ | $17.04_{\pm0.04}$ | $11.69_{\pm0.19}$ | | F | $19.00_{\pm1.39}$ | $19.66_{\pm1.52}$ | $18.37_{\pm1.09}$ |
| | Sp. | - | - | $98.54_{(76.94)}$ | | Sp. | - | - | $93.96_{(4.61)}$ |
| Ethanol | E | $0.32_{\pm0.04}$ | $0.34_{\pm0.01}$ | $0.21_{\pm0.01}$ | W | E | $3.10_{\pm0.01}$ | $2.27_{\pm0.06}$ | $1.93_{\pm0.18}$ |
| | F | $11.80_{\pm0.58}$ | $11.93_{\pm0.01}$ | $10.88_{\pm0.10}$ | | F | $51.98_{\pm0.89}$ | $54.54_{\pm1.71}$ | $52.86_{\pm1.58}$ |
| | Sp. | - | - | $98.40_{(74.81)}$ | | Sp. | - | - | $93.83_{(2.67)}$ |
| Naphthalene | E | $0.13_{\pm0.01}$ | $0.11_{\pm0.01}$ | $0.11_{\pm0.00}$ | Ag∪Au-PBE | E | $1.29_{\pm0.17}$ | $0.91_{\pm0.05}$ | $0.78_{\pm0.10}$ |
| | F | $9.87_{\pm0.30}$ | $8.47_{\pm0.80}$ | $8.70_{\pm0.48}$ | | F | $5.51_{\pm0.25}$ | $5.65_{\pm0.29}$ | $5.88_{\pm0.18}$ |
| | Sp. | - | - | $98.54_{(77.01)}$ | | Sp. | - | - | $94.39_{(11.49)}$ |
| Salicylic | E | $0.20_{\pm0.01}$ | $0.32_{\pm0.02}$ | $0.21_{\pm0.03}$ | H$_2$O-PD | E | $2.38_{\pm0.18}$ | $3.02_{\pm0.41}$ | $2.67_{\pm0.16}$ |
| | F | $14.16_{\pm0.17}$ | $14.43_{\pm0.58}$ | $13.07_{\pm0.51}$ | | F | $76.05_{\pm1.42}$ | $69.92_{\pm1.61}$ | $72.39_{\pm1.19}$ |
| | Sp. | - | - | $98.30_{(73.18)}$ | | Sp. | - | - | $93.76_{(1.48)}$ |
| Uracil | E | $0.18_{\pm0.02}$ | $0.37_{\pm0.06}$ | $0.30_{\pm0.10}$ | | | | | |
| | F | $11.85_{\pm0.18}$ | $16.16_{\pm4.19}$ | $12.50_{\pm0.14}$ | | | | | |
| | Sp. | - | - | $98.44_{(75.33)}$ | | | | | |

Table II: **Evaluation on the rMD17 datasets.** Energy MAE and force MAE values with standard deviation are reported in meV/atom and meV/Å, respectively. Sparsity values are reported as total sparsity with layer-wise sparsity in parentheses, in percent (%).

| Subset | Metric | Zero shot | Scratch | Method Full | ELoRA | Ours (L) | Ours (H) |
|---|---|---|---|---|---|---|---|
| Aspirin | Energy | 4.72 | $0.60_{\pm 0.07}$ | $0.19_{\pm 0.01}$ | $0.21_{\pm 0.01}$ | $0.17_{\pm 0.01}$ | $0.20_{\pm 0.02}$ |
| | Force | 354.42 | $25.55_{\pm 0.70}$ | $8.09_{\pm 0.09}$ | $8.52_{\pm 0.08}$ | $7.56_{\pm 0.08}$ | $8.22_{\pm 0.09}$ |
| | Sparsity | - | - | - | - | 83.19 (27.50) | 96.84 (86.38) |
| Benzene | Energy | 4.50 | $0.16_{\pm 0.07}$ | $0.02_{\pm 0.01}$ | $0.02_{\pm 0.00}$ | $0.01_{\pm 0.00}$ | $0.01_{\pm 0.00}$ |
| | Force | 233.45 | $7.63_{\pm 0.14}$ | $1.67_{\pm 0.08}$ | $0.94_{\pm 0.02}$ | $0.90_{\pm 0.04}$ | $0.96_{\pm 0.03}$ |
| | Sparsity | - | - | - | - | 83.50 (28.83) | 96.85 (86.40) |
| Malonaldehyde | Energy | 7.47 | $0.98_{\pm 0.12}$ | $0.20_{\pm 0.01}$ | $0.24_{\pm 0.01}$ | $0.21_{\pm 0.04}$ | $0.22_{\pm 0.01}$ |
| | Force | 405.37 | $22.66_{\pm 0.35}$ | $7.91_{\pm 0.09}$ | $8.61_{\pm 0.06}$ | $7.58_{\pm 0.13}$ | $8.03_{\pm 0.13}$ |
| | Sparsity | - | - | - | - | 83.54 (29.00) | 97.39 (88.73) |
| Paracetamol | Energy | 5.33 | $0.44_{\pm 0.04}$ | $0.12_{\pm 0.01}$ | $0.13_{\pm 0.01}$ | $0.11_{\pm 0.01}$ | $0.13_{\pm 0.01}$ |
| | Force | 232.60 | $21.39_{\pm 0.16}$ | $6.74_{\pm 0.01}$ | $6.65_{\pm 0.03}$ | $5.92_{\pm 0.05}$ | $6.53_{\pm 0.04}$ |
| | Sparsity | - | - | - | - | 83.29 (27.91) | 96.92 (86.72) |
| Toluene | Energy | 5.54 | $0.19_{\pm 0.00}$ | $0.05_{\pm 0.00}$ | $0.06_{\pm 0.00}$ | $0.05_{\pm 0.01}$ | $0.05_{\pm 0.00}$ |
| | Force | 280.54 | $11.78_{\pm 0.11}$ | $3.31_{\pm 0.12}$ | $3.01_{\pm 0.03}$ | $2.70_{\pm 0.02}$ | $2.92_{\pm 0.02}$ |
| | Sparsity | - | - | - | - | 83.31 (28.01) | 96.87 (86.51) |
| Azobenzene | Energy | 4.96 | $0.28_{\pm 0.02}$ | $0.10_{\pm 0.01}$ | $0.11_{\pm 0.01}$ | $0.08_{\pm 0.01}$ | $0.10_{\pm 0.01}$ |
| | Force | 353.55 | $17.05_{\pm 0.24}$ | $6.49_{\pm 0.07}$ | $6.59_{\pm 0.04}$ | $6.26_{\pm 0.59}$ | $6.90_{\pm 0.04}$ |
| | Sparsity | - | - | - | - | 83.20 (27.56) | 97.30 (88.36) |
| Ethanol | Energy | 6.93 | $0.33_{\pm 0.04}$ | $0.08_{\pm 0.01}$ | $0.10_{\pm 0.01}$ | $0.09_{\pm 0.01}$ | $0.10_{\pm 0.01}$ |
| | Force | 369.69 | $13.28_{\pm 0.60}$ | $4.24_{\pm 0.12}$ | $4.58_{\pm 0.09}$ | $4.12_{\pm 0.11}$ | $4.49_{\pm 0.10}$ |
| | Sparsity | - | - | - | - | 83.21 (27.58) | 96.32 (84.15) |
| Naphthalene | Energy | 5.63 | $0.19_{\pm 0.03}$ | $0.06_{\pm 0.00}$ | $0.07_{\pm 0.00}$ | $0.05_{\pm 0.01}$ | $0.06_{\pm 0.01}$ |
| | Force | 288.43 | $13.59_{\pm 0.17}$ | $3.75_{\pm 0.02}$ | $3.34_{\pm 0.03}$ | $3.25_{\pm 0.09}$ | $3.26_{\pm 0.04}$ |
| | Sparsity | - | - | - | - | 83.29 (27.91) | 97.33 (88.50) |
| Salicylic | Energy | 6.13 | $0.32_{\pm 0.01}$ | $0.09_{\pm 0.00}$ | $0.08_{\pm 0.00}$ | $0.08_{\pm 0.01}$ | $0.08_{\pm 0.00}$ |
| | Force | 351.31 | $20.87_{\pm 0.23}$ | $5.99_{\pm 0.07}$ | $5.87_{\pm 0.07}$ | $5.51_{\pm 0.18}$ | $5.72_{\pm 0.05}$ |
| | Sparsity | - | - | - | - | 83.05 (26.88) | 97.17 (87.78) |
| Uracil | Energy | 6.91 | $0.41_{\pm 0.09}$ | $0.08_{\pm 0.00}$ | $0.08_{\pm 0.01}$ | $0.07_{\pm 0.00}$ | $0.08_{\pm 0.01}$ |
| | Force | 365.86 | $17.41_{\pm 0.16}$ | $6.03_{\pm 0.15}$ | $4.96_{\pm 0.07}$ | $4.52_{\pm 0.21}$ | $4.92_{\pm 0.07}$ |
| | Sparsity | - | - | - | - | 83.69 (29.63) | 97.44 (88.95) |

Table III: **Evaluation on the LAM benchmark datasets.** Energy MAE and force MAE values with standard deviation are reported in meV/atom and meV/Å, respectively. Sparsity values are reported as total sparsity with layer-wise sparsity in parentheses, in percent (%).

| Subset | Metric | Method | | | | | |
|---|---|---|---|---|---|---|---|
| | | Zero shot | Scratch | Full | ELoRA | Ours (L) | Ours (H) |
| Al∪Mg∪Cu | Energy | 238.42 | $27.90_{\pm 4.22}$ | $1.33_{\pm 0.14}$ | $1.63_{\pm 0.12}$ | $1.83_{\pm 0.22}$ | $2.23_{\pm 0.21}$ |
| | Force | 43.17 | $8.23_{\pm 0.71}$ | $5.59_{\pm 0.03}$ | $6.82_{\pm 0.08}$ | $6.10_{\pm 0.07}$ | $6.63_{\pm 0.11}$ |
| | Sparsity | - | - | - | - | 97.61 (36.59) | 99.62 (89.98) |
| Cu | Energy | 359.05 | $3.31_{\pm 1.20}$ | $0.66_{\pm 0.01}$ | $0.79_{\pm 0.01}$ | $0.65_{\pm 0.02}$ | $0.71_{\pm 0.04}$ |
| | Force | 50.94 | $7.08_{\pm 0.72}$ | $3.96_{\pm 0.00}$ | $5.29_{\pm 0.02}$ | $4.32_{\pm 0.02}$ | $4.80_{\pm 0.02}$ |
| | Sparsity | - | - | - | - | 97.57 (35.38) | 99.58 (88.82) |
| Sn | Energy | 31905.00 | $8.60_{\pm 1.53}$ | $32.74_{\pm 1.36}$ | $9.33_{\pm 0.30}$ | $2.18_{\pm 0.01}$ | $2.49_{\pm 0.05}$ |
| | Force | 114.22 | $29.58_{\pm 0.72}$ | $25.82_{\pm 0.53}$ | $32.46_{\pm 0.64}$ | $23.92_{\pm 0.40}$ | $25.39_{\pm 0.47}$ |
| | Sparsity | - | - | - | - | 97.46 (32.59) | 99.42 (84.72) |
| SSE-PBE | Energy | 20.33 | $0.53_{\pm 0.03}$ | $0.29_{\pm 0.02}$ | $0.41_{\pm 0.01}$ | $0.33_{\pm 0.03}$ | $0.32_{\pm 0.01}$ |
| | Force | 82.38 | $12.74_{\pm 0.04}$ | $8.13_{\pm 0.18}$ | $10.85_{\pm 0.17}$ | $8.92_{\pm 0.28}$ | $9.78_{\pm 0.17}$ |
| | Sparsity | - | - | - | - | 96.77 (14.16) | 99.33 (82.12) |
| Ti | Energy | 126.38 | $8.61_{\pm 0.45}$ | $2.57_{\pm 0.22}$ | $3.50_{\pm 0.23}$ | $2.59_{\pm 0.16}$ | $3.28_{\pm 0.33}$ |
| | Force | 133.56 | $51.38_{\pm 3.26}$ | $39.24_{\pm 1.96}$ | $43.65_{\pm 2.60}$ | $39.01_{\pm 1.68}$ | $41.03_{\pm 1.90}$ |
| | Sparsity | - | - | - | - | 97.23 (26.35) | 99.45 (85.47) |
| V | Energy | 226.44 | $12.09_{\pm 0.81}$ | $1.45_{\pm 0.11}$ | $2.65_{\pm 0.16}$ | $1.67_{\pm 0.09}$ | $3.16_{\pm 0.47}$ |
| | Force | 168.02 | $41.26_{\pm 2.94}$ | $23.15_{\pm 1.69}$ | $31.25_{\pm 1.83}$ | $25.38_{\pm 1.50}$ | $29.26_{\pm 1.80}$ |
| | Sparsity | - | - | - | - | 97.60 (36.28) | 99.65 (90.74) |
| W | Energy | 378.04 | $7.22_{\pm 0.92}$ | $2.94_{\pm 0.10}$ | $4.33_{\pm 0.52}$ | $3.18_{\pm 0.09}$ | $4.03_{\pm 0.17}$ |
| | Force | 449.23 | $96.13_{\pm 2.74}$ | $66.00_{\pm 2.28}$ | $89.26_{\pm 2.35}$ | $71.71_{\pm 2.11}$ | $79.01_{\pm 2.46}$ |
| | Sparsity | - | - | - | - | 97.44 (32.04) | 99.45 (85.33) |
| Ag∪Au-PBE | Energy | 80.15 | $14.27_{\pm 0.91}$ | $3.64_{\pm 0.36}$ | $1.89_{\pm 0.19}$ | $1.26_{\pm 0.13}$ | $1.40_{\pm 0.16}$ |
| | Force | 93.36 | $17.55_{\pm 2.53}$ | $14.03_{\pm 0.93}$ | $10.39_{\pm 0.74}$ | $8.55_{\pm 0.54}$ | $9.24_{\pm 0.61}$ |
| | Sparsity | - | - | - | - | 97.38 (30.34) | 99.41 (84.22) |
| $H_2O$-PD | Energy | 311.19 | $5.10_{\pm 0.89}$ | $11.67_{\pm 0.38}$ | $4.98_{\pm 0.41}$ | $3.67_{\pm 0.23}$ | $4.84_{\pm 0.35}$ |
| | Force | 346.92 | $80.64_{\pm 8.12}$ | $84.25_{\pm 1.67}$ | $69.41_{\pm 3.24}$ | $62.31_{\pm 1.52}$ | $67.71_{\pm 1.65}$ |
| | Sparsity | - | - | - | - | 97.47 (32.82) | 99.48 (86.30) |

