# OpenReview forum: "Robust and Interpretable Adaptation of Equivariant Materials Foundation Models via Sparsity-promoting Fine-tuning"
_ICLR.cc/2026/Conference — ICLR 2026 Poster_

### Official Review · Reviewer_cUD3 · 2025-11-01

**Soundness:** 3
**Presentation:** 3
**Contribution:** 3
**Rating:** 4
**Confidence:** 3

**Summary:**

The authors present a method that enables pre-trained equivariant models for materials and molecules to be fine-tuned for new downstream tasks (e.g. different level of theory, different elemental compositions, etc) with very minimal parameters updates (< 5%) while maintaining equivariance. They use soft threshold weight reparameterization (STR) to learn the adaptation weights in a manner that promotes sparseness, so the original weights are only updated when necessary. Results are presented on LAM, MD17, TM-O-Spin, and MP-mag, in nearly all cases the method is reported to be equivalent or better than full fine-tuning.

**Strengths:**

- The work addresses an important and timely topic.
- A notable strength is the exceptional parameter efficiency: the model adapts to new tasks with surprisingly few parameter updates while maintaining good performance.
- The experimental design is rigorous and well-motivated, with appropriate baselines and comprehensive ablation studies.

**Weaknesses:**

- While reporting the fraction of parameters updated is great, it doesn’t help understand what the practical benefits of the methods are i.e. how much less memory is required?, how much faster is training than full fine-tuning?, can you store different adopters for different tasks?, etc. It would make the paper much stronger with this information included and emphasized.
- The paper does not discuss in-depth why this fine-tuning method is consistently more accurate than full fine-tuning. This seems like a counter initiative result and is worth digging into more.
- In particular, the MD17 benchmark is quite low signal i.e. small tweaks in hyperparameters can change the outcome.

**Questions:**

1. How much hyperparameter tuning was done for your method vs that done for full fine-tuning or ELoRA?
2. Was the schedule-free AdamW optimizer used for all experiments?
3. The percentage of parameters that get updated is impressively small, is there precedent for this (or examples of this) in other fields?
4. Are the adaptation weights just added to the original weights? Or is something else happening?
5. I was confused by line 168, does the method only adapt path weights on scalar features (ignoring higher order features) or is it saying that path weights are scalar values? Also, what weights are being adapted in the linear layers?
6. What are the limitations of the method?
7. Quick and easy fine-tuning is great to have, but as the foundational models get bigger distillation will also be important as well. Does any of the work here carry over to distillation?
8. The S in the table 1. makes me think of stress not sparsity.

---

> ### Author Response · Authors · 2025-11-24
> **Response 1/3**
>
> We greatly appreciate the reviewer’s thoughtful evaluation. We have provided detailed point-by-point responses to all comments raised.
>
> > __W1-1. Questions regarding computational efficiency__
>
> First, we clarify that the objective of our method is not to reduce computational cost or memory usage during fine-tuning, and we do not claim such benefits in the paper. The main purpose of our work is to induce sparsity in $\Delta W$, thereby improving adaptation accuracy and providing interpretability as demonstrated in __Figure 2__. In the revised manuscript, we have clarified that the objective of our method is not to reduce computational costs __(Lines 172-175)__.
>
> In response to the reviewer’s question, we measured the computational cost during model fine-tuning. Following our experimental setup, we fine-tuned the MACE-OFF23-medium model on the rMD17 Aspirin dataset using a single NVIDIA H100 GPU and measured the training iteration time and peak memory usage. The results in the table below show that our method incurs overhead compared to full fine-tuning in both computation time and memory usage, but the overhead is marginal. Detailed experimental settings and result analysis are provided in __Section 5.5__ of the revised manuscript.
>
> |     Method    | Avg. time (s) | Memory (MB) |
> |:-------------:|:-------------:|:-----------:|
> | Full          |    0.14492    |    5291.7   |
> | ELoRA (r=16)  |    0.16556    |    5805.2   |
> | Ours (Linear) |    0.15949    |    5302.0   |
> | Ours (FCTP)   |    0.14977    |    5828.3   |
> | Ours (All)    |    0.16175    |    5834.3   |
>
> > __W1-2. “Can you store different adopters for different tasks?”__
>
> Yes, our method allows storing different $\Delta W$ for different tasks. In this case, the $\Delta W$ can be stored in sparse matrix format, making it more storage-efficient than full fine-tuning.
>
> > __W2. “The paper does not discuss in-depth why this fine-tuning method is consistently more accurate than full fine-tuning.”__
>
> Our approach explicitly limits the number of parameters updated during fine-tuning, typically modifying less than 3 % in inorganic datasets and less than 17 % in organic datasets of the pretrained MACE models. __This rigorous selection significantly reduces the optimization search space compared to full fine-tuning, thereby stabilizing training and improving generalization and accuracy when data is limited.__
>
> This strategy is supported by findings in the NLP domain, which indicate that pretrained language models rely on a small subset of critical parameters for adaptation. Gong et al. [1] reported that sparse fine-tuning across different downstream tasks exhibits highly similar sparsity patterns, suggesting the existence of task-agnostic sparse updates that represent ‘winning lottery tickets’ within pretrained language models. Song et al. [2] observed that during full fine-tuning, strong gradients flow through only a small fraction of parameters while the majority remain relatively unchanged, demonstrating a ‘quasi-sparse’ update pattern. Together, these observations support the view that adjusting a small number of critical parameters is typically sufficient for downstream adaptation. __We conjecture that similar principles apply to MLIPs, given the shared nature of adapting pretrained models to downstream tasks, where forcing the sparsity helps narrow the parameter space to update.__
>
> Finally, we note that the performance gap among full fine-tuning, ELoRA, and our method naturally tends to narrow as the amount and diversity of fine-tuning data increase, as discussed in __Section 5.3__ of our paper and __Figure 4__ of the ELoRA paper [3]. This observation is consistent with the understanding that the benefits of implicit regularization are most pronounced when fine-tuning data is limited, and diminish as more abundant training data becomes available.
>
>
> __(Continued in the following response)__

---

> ### Author Response · Authors · 2025-11-24
> **Response 2/3**
>
> > __W3 & Q1. “the MD17 benchmark is quite low signal” & “How much hyperparameter tuning was done…?”__
>
> We acknowledge the reviewer’s valid concern regarding the ‘low signal’ nature of the original MD17 benchmark. As noted by Christensen and Lilienfeld [4], the original MD17 dataset [5] suffers from noisy ground truth labels. To mitigate this issue, we instead employed the revised MD17 (rMD17) dataset [4], which provides reliable energies and forces derived from tighter quantum-mechanical calculations.
>
> Regarding the extent of hyperparameter tuning, we followed the optimal settings reported in the ELoRA paper [3] for full fine-tuning and ELoRA to ensure a fair and rigorous comparison. Specifically, we utilized the finalized hyperparameters described in __Table 6 of [3]__, which were determined through their comprehensive ablation studies __(Figures 5 and 7 in [3])__, to ensure best configurations.
>
> For our method, we established the initial search range based on ELoRA’s configuration. We then conducted focused ablation studies for the learning rate and the threshold value $\tau$, as detailed in __Figure 1__ and __Appendix Figure I__ of our manuscript. As presented in the table below, our ablation results on the rMD17 Aspirin dataset show that a learning rate of $1\times 10^{-2}$ demonstrates the optimal accuracy for both energy and force predictions; consequently, we used this setting across all experiments.
>
> | Learning rate |  5e-2 | 2e-2  | 1e-2  | 5e-3  | 2e-3  | 1e-3  | 5e-4  |
> |:--:|:-----:|-------|-------|-------|-------|-------|-------|
> | Energy  |  0.18 |  0.19 |  0.17 |  0.19 |  0.19 |  0.22 |  0.26 |
> | Force  |  7.65 |  7.61 |  7.56 |  7.58 |  7.87 |  8.34 |  9.35 |
> | Total Sparsity | 83.11 | 82.90 | 83.19 | 83.08 | 82.69 | 81.13 | 79.40 |
>
> In the revised manuscript, we provide a clarification on how the MD17 and rMD17 datasets differ from each other __(Lines 220–222)__, and add a detailed explanation of the hyperparameter setting for each method in __Section 4.2__.
>
> > ___Q2. “Was the schedule-free AdamW optimizer used for all experiments?”___
>
> Yes. We used the schedule-free AdamW optimizer [6] consistently across all experiments reported in the paper.
>
> > __Q3. “The percentage of parameters that get updated is impressively small, is there precedent for this (or examples of this) in other fields?”__
>
> Yes. As noted in our response to W2, there are clear precedents in the NLP domain, where sparsity has been introduced for fine-tuning language models. For example, Gong et al. [1] fine-tuned a pretrained language model by updating only about 0.05 % of the parameters while still achieving strong performance. Similarly, Song et al. [2] proposed SIFT, which updates roughly 0.2 % of the parameters and also obtained competitive results.
>
> > ___Q4. “Are the adaptation weights just added to the original weights? Or is something else happening?”___
>
> The interaction between the original weights and the adaptation weights follows the same mechanism as vanilla LoRA [7], with a slight distinction between training and inference.
>
> During training, features are computed separately using the original weight $W$ and the adaptation weight $\Delta W$, and the resulting features are then summed.
> During inference, the two weight tensors are directly added to obtain the final weight $W+\Delta W$, which is subsequently used to compute the features.
>
>
> __(Continued in the following response)__

---

> ### Author Response · Authors · 2025-11-24
> **Response 3/3**
>
> >__Q5. I was confused by line 168, does the method only adapt path weights on scalar features (ignoring higher order features) or is it saying that path weights are scalar values? Also, what weights are being adapted in the linear layers?__
>
> We apologize for the ambiguity in Line 168. To answer the reviewer’s question directly, the learnable path weights are scalar values, and they scale the interaction paths for both scalar and higher-order tensor features.
>
> All tensor product operations in equivariant models follow the Clebsch–Gordan rule to ensure equivariance. The channel-wise scalar Clebsch–Gordan coefficients used in each interaction path are pre-defined by the theory (as described in __Appendix A__ and __Equation 5__). Our fine-tuning approach introduces an update $\Delta W$ that acts on these scalar path weights in the equivariant layers (including linear ones), without altering the underlying Clebsch–Gordan product structure, thereby maintaining equivariance.
>
> In the revised manuscript, we have clarified this mechanism to ensure the distinction and the preservation of equivariance are explicitly stated __(Lines 168-169)__.
>
> >__Q6. “What are the limitations of the method?”__
>
> Generally, sparse neural networks are often studied with the goal of achieving computational efficiency. However, in the fine-tuning scenario, the original weights remain dense, and consequently, the final weights are also dense, preventing us from obtaining computational advantages. As we mentioned in the ‘Outlook’ section, we believe that structured sparse pretraining represents a promising research direction to address this limitation.
>
> >__Q7. “Does any of the work here carry over to distillation?”__
>
> Our method induces sparsity in the updated weights during fine-tuning. While our work focuses on fine-tuning rather than distillation, the idea of enforcing sparsity itself could extend to distillation settings. For example, one could train a student model in a sparse manner, which aligns conceptually with the sparse pretraining.
>
> >__Q8. “The S in the table 1. makes me think of stress not sparsity.”__
>
> We appreciate this suggestion. To avoid confusion with ‘stress’, we have changed the notation for sparsity from ‘S’ to ‘Sp.’ in __Table 1__ of the revised manuscript.
>
> __Reference__
>
> [1] Gong et al., “Finding the Dominant Winning Ticket in Pre-Trained Language Models.” ACL (2022).
>
> [2] Song et al., “Sparse is Enough in Fine-tuning Pre-trained Large Language Models.” ICML (2024).
>
> [3] Wang et al., “ELoRA: Low-Rank Adaptation for Equivariant GNNs.” ICML (2025).
>
> [4] Christensen and Lilienfeld, “On the role of gradients for machine learning of molecular energies and forces.” Machine Learning: Science and Technology. 1, 045018 (2020).
>
> [5] Chmiela et al., “Machine learning of accurate energy-conserving molecular force fields.” Science Advances. 3(5), e1603015 (2017).
>
> [6] Defazio et al., “The Road Less Scheduled.” NeurIPS (2024).
>
> [7] Hu et al., “LoRA: Low-Rank Adaptation of Large Language Models.” ICLR (2022).

---

### Official Review · Reviewer_bcJ7 · 2025-11-01

**Soundness:** 2
**Presentation:** 3
**Contribution:** 2
**Rating:** 4
**Confidence:** 4

**Summary:**

The paper proposes a sparsity-promoting finetuning method for E(3)-equivariant MLIPs (e.g., MACE). The authors enforce the finetuned weights ΔW as a sparse format by using a Soft Threshold Reparameterization (STR) method, which only updates a small subset of interaction paths. This method can preserve equivariance during adaptation. Experiments on rMD17 (molecules), LAM subsets (crystals), and two magnetic datasets (TM-O-Spin, MP-mag) show that this sparse fine-tuning method can match or outperform full updating and ELoRA while only 0.5% to 3% of parameters are updated.

**Strengths:**

Employing STR with a learnable threshold per layer and decoupled optimization for ΔW vs. τ is simple, and the implementation appears lightweight.
Experimental Results: This paper covers molecules, multiple crystal subsets, and magnetic datasets. The results show that both low and high sparsity scenarios (meaning very few updated parameters), sparse finetuning gets lower error than full finetuning and ELoRA.

**Weaknesses:**

Evaluation: Sparse training is emphasized, but the training/inference time speed/footprint gains (if any) aren’t measured (e.g., wall-clock per step, peak memory usage). In Eq. (2), ΔW is initialized from a normal distribution and then sparsified via STR. I suspect this operation will not reduce training time and memory. Please provide memory profiles vs full finetuning and ELoRA.
Limited Benchmarks: The experiments mainly focus on MACE. It would be better to include Nequip-OAM-L to demonstrate the effectiveness of the sparse finetuning method.

**Questions:**

On the reproducibility of rMD17 dataset: In MACE: Higher Order Equivariant Message Passing Neural Networks for Fast and Accurate Force Fields (Table 1), the aspirin setting reports Etot=2.2 meV (original metric), whereas Table 1 in the author’s paper reports 0.6 meV/atom. Please provide a MACE-aligned reproduction (mean±std over seeds) and, ideally, attach the exact training config/script in the rebuttal for community verification.

The baseline results appear inconsistent with those reported in prior work. For instance, in the case of aspirin, MACE [1] reports errors of 2.2 meV for energy and 6.6 meV/Å for forces when trained from scratch. In contrast, this paper reports 0.60 meV/atom (equivalent to ~12.6 meV per molecule) and 25.55 meV/Å. Such a significant discrepancy raises the question of whether the baseline models in this study were suboptimally configured or tuned. It is recommended that the authors either realign their baselines with established reference values or provide a clear explanation for these differences.

---

> ### Author Response · Authors · 2025-11-24
> **Response 1/2**
>
> We are grateful to the reviewer for the thorough and constructive review. We have addressed all weaknesses and questions point-by-point in the following response.
>
> > __W1 & W2. Questions regarding computational efficiency__
>
> We first clarify that the objective of our method is not to reduce computational cost or memory usage during training, and we do not claim such benefits in the paper. Rather, we introduce sparsity in the weight update during fine-tuning, which offers a distinct advantage of physical interpretability as shown in __Figure 2__. In the fine-tuning scenario of our work, as the original parameters $W$ remain dense, inducing sparsity only in $\Delta W$ does not inherently reduce computational cost during training. At inference time, the cost remains identical to the original model, as the two weight tensors are directly added to obtain the final weight $W+\Delta W$ before computing features.
>
> In response to the reviewer’s request, we measured the training iteration time and memory consumption of full fine-tuning, ELoRA [5], and our method. To align with our main experiment, we fine-tuned the MACE-OFF23-medium model [4] using the rMD17 Aspirin dataset and measured the average iteration time and memory usage. We performed 20 warm-up iterations and measured the average over the following 20 iterations, repeated three times, with our default batch size (64) on a single NVIDIA H100 GPU.
>
> |     Method    | Avg. time (s) | Memory (MB) |
> |:-------------:|:-------------:|:-----------:|
> | Full          |    0.14492    |    5291.7   |
> | ELoRA (r=16)  |    0.16556    |    5805.2   |
> | Ours (Linear) |    0.15949    |    5302.0   |
> | Ours (FCTP)   |    0.14977    |    5828.3   |
> | Ours (All)    |    0.16175    |    5834.3   |
>
> __Training time__: Our proposed methods introduce only a modest computational overhead, generally ranging from 4 % to 7 %, slower than full fine-tuning. Notably, all our settings are consistently faster than ELoRA. This advantage stems from the matrix multiplication required in ELoRA to reconstruct $\Delta W$, which becomes a bottleneck in the GPU.
>
> __Memory usage__: The memory overhead of our ‘Linear’ setting is marginal compared to full fine-tuning. When using the ‘All’ setting of our method, the additional memory cost is comparable to ELoRA. Unlike LLMs, MLIPs do not generally enjoy significant memory savings, as memory usage is dominated by the intermediate activations and gradients required for autograd-based force prediction, rather than by the optimizer state size.
>
> We have clarified in the revised manuscript that the objective of our method is not to reduce computational costs __(Lines 172–175)__, and we have added __Section 5.5__ to report and analyze the measured computational cost and memory usage.
>
>
> __(Continued in the following response)__

---

> ### Author Response · Authors · 2025-11-24
> **Response 2/2**
>
> > __W3. Limited benchmarks: Experiments on Nequip-OAM-L__
>
> Following the reviewer’s suggestion, __we provide additional results comparing full fine-tuning and our (linear) method on the NequIP-OAM-L model [1], as detailed in Appendix D and Appendix Table III.__ We obtained initial results on three subsets from both the rMD17 and the LAM benchmarks, observing that our method achieves performance comparable to full fine-tuning. We note that these results were obtained under preliminary hyperparameter settings, and further optimization may improve the overall performance of both methods. We thank the reviewer’s suggestion, and will include NequIP results in the final manuscript.
>
> > __Q1. Discrepancy between our paper and MACE paper__
>
> We appreciate the reviewer's careful examination. __We clarify that the reproducibility issue regarding our results and the NeurIPS 2022 paper [2] referenced by the reviewer arises from using fundamentally different MACE backbones, not suboptimal tuning.__
>
> The NeurIPS paper [2] utilized the original high-capacity MACE architecture with hidden_irreps = ‘256x0e + 256x1o + 256x2e’ (see [3]) specifically trained for each task. In contrast, we employ MACE-OFF23 [4], a molecular foundation model within the MACE family, which uses hidden_irreps = ‘128x0e + 128x1o’. The ‘hidden_irreps’ describe the irreducible-representation-structured internal node-feature space: the original configuration ('256x0e + 256x1o + 256x2e') corresponds to a total of 2,304 channels, whereas the compact MACE-OFF23 configuration (‘128x0e + 128x1o’) provides 512 channels. This represents an effective fourfold reduction in capacity, reflecting an intentional architectural choice in MACE-OFF23 that improves computational efficiency while remaining sufficient for organic systems governed primarily by $s$- and $p$-orbitals.
>
> Additionally, we clarify that our research objective is to evaluate the fine-tuning methods for foundation MLIPs. Therefore, __to ensure a fair comparison, all approaches must begin with the same pretrained foundation model.__ Since MACE-OFF23 is a molecular foundation model, fine-tuning it on rMD17 (which was unseen during pretraining) provides a natural experimental setting for molecular systems. Similarly, for LAM experiments, we utilize MACE-MP-0b3 [5], the corresponding inorganic crystal foundation model.
>
> Furthermore, our experimental setup aligns with prior work such as ELoRA [6], which also utilizes MACE-OFF and MACE-MP as base models. Because all fine-tuning experiments in our study are built upon MACE-OFF and MACE-MP architectures, we also use these architectures for our ‘from scratch’ baseline to ensure consistency in model capacity and architectural assumptions across all comparisons.
>
> In summary, the numerical differences between the NeurIPS paper [2] and our paper are a natural consequence of distinct backbone architectures and experimental context, and not from suboptimal tuning. Our experimental design ensures rigorous and fair comparisons across fine-tuning methods while aligning with current practices in foundation MLIP research.
>
> We appreciate the reviewer’s constructive feedback, and we have clarified this point in __Section 4.2__ of the revised manuscript __(Lines 245-248)__.
>
> __Reference__
>
> [1] Batzner et al., “E(3)-equivariant graph neural networks for data-efficient and accurate interatomic potentials.” Nature Communications 13, 1, 2453 (2022).
>
> [2] Batatia et al., “MACE: Higher Order Equivariant Message Passing Neural Networks for Fast and Accurate Force Fields.” NeurIPS (2022).
>
> [3] https://github.com/ACEsuit/mace/discussions/165
>
> [4] Kovács et al., “MACE-OFF: Short-Range Transferable Machine Learning Force Fields for Organic Molecules.” Journal of the American Chemical Society, 147, 21, 17598–17611 (2025).
>
> [5] Batatia et al., “A foundation model for atomistic materials chemistry.” The Journal of Chemical Physics, 163, 184110 (2025).
>
> [6] Wang et al., “ELoRA: Low-Rank Adaptation for Equivariant GNNs.” ICML (2025).

---

### Official Review · Reviewer_KPDY · 2025-11-01

**Soundness:** 2
**Presentation:** 3
**Contribution:** 2
**Rating:** 4
**Confidence:** 2

**Summary:**

This paper proposes a sparsity-promoting fine-tuning method for equivariant materials foundation models, using Soft Threshold Weight Reparameterization (STR) to reduce the number of updated parameters. On molecular and crystalline benchmarks, it updates only a small fraction of parameters while matching or surpassing full fine-tuning.

**Strengths:**

+Strong fine-tuning accuracy across molecules, inorganic crystals, and magnetic tasks.
+Extends force-field foundation models to magnetic systems for magnetic moment prediction.
+Provides a novel analysis linking valence electronic structure with sparse update patterns.

**Weaknesses:**

-Limited novelty: mainly applying STR to equivariant models.
-Experimental results are not aligned with prior work and inconsistent across sections.
-The interpretability analysis is not fully convincing.

**Questions:**

1. STR is applied to \Delta W rather than W (Sec. 3.2). Does this double the number of parameters during training? Without structured sparsity, are these computations still dense?
2. In Eq. (3), the mask uses \Delta W > 0, which blocks gradients for negative weights. Is this a typo or an intentional design choice?
3. The definition of sparsity is confusing. Lines 316-317 state “Total sparsity measures the fraction of updated parameters,” implying that higher sparsity means more updates. Yet Table 1 shows 80-100\% “sparsity,” while the abstract claims only 0.5-3\% updated. Please unify the definition.
4. Baselines are misaligned with prior work. Taking aspirin as an example, MACE [1] reports 2.2 meV (energy) and 6.6 meV/Å (force) from scratch, while this paper reports 0.60 meV/atom (12.6 meV) and 25.55 meV/Å. Does this indicate suboptimal tuning for baselines? Please align baselines or explain the discrepancies.
5. TM-O-Spin: Appendix Fig. I shows force MAE ≥ 70 meV/Å in ablations, but Table 2 reports 48.75 meV/Å for Ours (L). Why the mismatch?
6. Interpretability: Why does ELoRA show updates for elements outside the training set? With one-hot element embeddings in MACE, absent elements should not activate or receive gradients. Official MACE implementations typically remove such parameters before fine-tuning.
[1] Batatia, I., Kovacs, D. P., Simm, G., Ortner, C., & Csányi, G. (2022). MACE: Higher order equivariant message passing neural networks for fast and accurate force fields. Advances in neural information processing systems, 35, 11423-11436.

---

> ### Author Response · Authors · 2025-11-24
> **Response 1/3**
>
> We thank reviewer KPDY for the valuable and supportive feedback. We have thoroughly reviewed and revised the manuscript to enhance its overall rigor and presentation.
>
> > __W1. Limited novelty: mainly applying STR to equivariant models__
>
> __Our core novelty lies in establishing sparsity as a viable and beneficial paradigm for fine-tuning equivariant MLIPs__, a direction that has not been explored before despite its significance. This novelty emerges from both a technical contribution and a domain-specific synergy uniquely enabled by sparsity in equivariant MLIPs.
>
> __[In terms of technical novelty]__: Incorporating sparsity into these highly constrained, equivariant architectures is non-trivial. Naive sparsity-promoting methods, such as $L_0$ regularization [1], or even vanilla STR [2] itself, suffered from severe instability in our preliminary experiments. This instability arises because standard STR enforces a shared weight decay for $\tau$ and $\Delta W$, preventing the precise calibration required for stable sparsification via $\tau$. Consequently, even with a small ($\leq 1\times 10^{-4}$) weight decay triggers coupled oscillations, where rapid shifts in the threshold alter the pruning mask and thus destabilize weight updates. __Our key contribution is the decoupling of the weight decay for $\tau$ and $\Delta W$ (Equations 2 and 3), which enables independent control over sparsification dynamics.__ This critical refinement ensures training stability and precise sparsity control, distinguishing our work from a trivial application of existing techniques.
>
> __[In terms of domain synergy]__: Sparsity produces unique and meaningful synergies with the structure of equivariant MLIPs. Since the parameters in these models correspond to physically meaningful components (e.g., spherical harmonics), updating only a subset of these elements through sparse fine-tuning enhances interpretability. This __enables identification of which physical components are most relevant for a given task__, as illustrated in Figure 2 of our manuscript.
>
> In summary, our work opens a previously unexplored direction in the fine-tuning of equivariant MLIPs, demonstrates domain-specific advantages that emerge from the interaction between sparsity and physical structure, and provides a technical approach to realize these benefits in practice.
>
> > __W2 & Q5. “TM-O-Spin: Appendix Fig. I shows force MAE ≥ 70 meV/Å in ablations, but Table 2 reports 48.75 meV/Å for Ours (L). Why the mismatch?”__
>
> We sincerely thank the reviewer for pointing out this discrepancy and are grateful for careful attention to detail. Upon re-examining the reported results, we found that 70.46 meV/Å in __Appendix Figure I__ is correct, and we corrected the value in __Table 2__ accordingly. We note that this correction does not affect the overall results or trends reported in the paper.
>
> __(Continued in the following response)__

---

> ### Author Response · Authors · 2025-11-24
> **Response 2/3**
>
> > __W2 & Q4. Discrepancy between our paper and MACE paper__
>
> We appreciate the reviewer's careful examination. __The discrepancies in values between our paper and the NeurIPS 2022 paper [3] referenced by the reviewer arise from fundamental architectural differences in the MACE backbones employed, not from suboptimal tuning.__
>
> The NeurIPS paper [3] proposed the original high-capacity MACE architecture with hidden_irreps = ‘256x0e + 256x1o + 256x2e’ (see [4]), and the model was specifically trained for each task. In contrast, we employ MACE-OFF23 [5], a molecular foundation model within the MACE family, which uses hidden_irreps = ‘128x0e + 128x1o’. The ‘hidden_irreps’ describe the irreducible-representation-structured internal node-feature space: the original configuration ('256x0e + 256x1o + 256x2e') corresponds to a total of 2,304 channels, whereas the compact MACE-OFF23 configuration (‘128x0e + 128x1o’) provides 512 channels. This results in an effective fourfold reduction in capacity, which is an intentional architectural choice of MACE-OFF23 to improve computational efficiency while remaining sufficient for organic systems governed primarily by $s$- and $p$-orbitals.
>
> In addition, we clarify that our research objective is to evaluate the fine-tuning methods for foundation MLIPs. Therefore, __to ensure a fair comparison, all approaches must begin with the same pretrained foundation model.__ Since MACE-OFF23 is a molecular foundation model, fine-tuning it on rMD17 (which was unseen during pretraining) provides a natural experimental setting for molecular systems. Similarly, for LAM experiments, we utilize MACE-MP-0b3 [6], the corresponding inorganic crystal foundation model.
>
> Furthermore, our experimental setup aligns with prior work such as ELoRA [7], which also utilizes MACE-OFF and MACE-MP as base models. Because all fine-tuning experiments in our study are built upon MACE-OFF and MACE-MP architectures, we also use these architectures for our ‘from scratch’ baseline to ensure consistency in model capacity and architectural assumptions across all comparisons.
>
> In summary, the numerical differences between the NeurIPS paper [3] and our paper are a natural consequence of distinct backbone architectures and experimental context, and not from suboptimal tuning. Our experimental design ensures rigorous and fair comparisons across fine-tuning methods while aligning with current practices in foundation MLIP research.
>
> We appreciate the reviewer’s helpful feedback, and we have clarified this point in __Section 4.2__ of the revised manuscript __(Lines 245-248)__.
>
>
>
>
>
> > __W3 & Q6. Questions regarding model interpretation__
>
> We understand the reviewer's point to be: __“Why does ELoRA update parameters for atomic species not present in the fine-tuning data?”__
>
> As the reviewer correctly pointed out, parameters corresponding to atomic species absent from the training data should not receive gradients, and indeed they do not. However, they are still updated due to the inherent structure of the low-rank parameterization.
>
> ELoRA represents the weight update $\Delta W$ as the product of two low-rank matrices, $A$ and $B$. Because $\Delta W=AB$, any parameter update applied to $A$ or $B$ propagates across entire rows or columns of $\Delta W$. Specifically, updating a single row of $A$ affects all columns of $\Delta W$ that depend on that row, and similarly, updating a column of $B$ affects all rows of $\Delta W$. Consequently, modifying a single entry of $\Delta W$ cannot be done independently; it is intrinsically coupled with other entries through the shared low-rank factors in $A$ and $B$. In other words, __when gradients from atomic species in the training data update specific rows or columns of $A$ and $B$, the low-rank structure propagates these changes across $\Delta W$__, indirectly influencing parameters corresponding to unseen atomic species through the shared factors.
>
> As shown in __Figure 2__ in our manuscript, the blurry update patterns in ELoRA arise directly from this reason, as the shared low-rank factors cause changes to spread across all parameters. This demonstrates that our method, which updates parameters independently and sparsely, is more effective in terms of interpretability. Specifically, for the TM-O-Spin dataset, ELoRA’s blurry background pattern hinders clear interpretation, whereas our method shows updates concentrated precisely on the atomic species present in the training data.
>
> We have incorporated the above explanation in __Section 5.4__ of the revised manuscript, and we hope it addresses the reviewer’s concern more clearly.
>
> __(Continued in the following response)__

---

> ### Author Response · Authors · 2025-11-24
> **Response 3/3**
>
> > __Q1. STR is applied to $\Delta W$ rather than $W$ (Sec. 3.2). Does this double the number of parameters during training? Without structured sparsity, are these computations still dense?__
>
> Since we do not introduce structured sparsity, the computations remain dense. During training, each targeted equivariant layer stores both the original weight $W$ and $\Delta W$, thereby doubling the parameter count. However, because we only fine-tune selected equivariant layers rather than the entire model, the total parameter count is not doubled.
>
> To provide more detailed information, we measured the computational cost during model fine-tuning. Following our experimental setup, we fine-tuned the MACE-OFF23-medium model on the rMD17 Aspirin dataset using a single NVIDIA H100 GPU and measured the training iteration time and peak memory usage. The results in the table below show that our method incurs overhead compared to full fine-tuning in both computation time and memory usage, but the overhead is marginal. Detailed experimental settings and result analysis are provided in __Section 5.5__ of the revised manuscript.
>
> |     Method    | Avg. time (s) | Memory (MB) |
> |:-------------:|:-------------:|:-----------:|
> | Full          |    0.14492    |    5291.7   |
> | ELoRA (r=16)  |    0.16556    |    5805.2   |
> | Ours (Linear) |    0.15949    |    5302.0   |
> | Ours (FCTP)   |    0.14977    |    5828.3   |
> | Ours (All)    |    0.16175    |    5834.3   |
>
> > __Q2. In Eq. (3), the mask uses $\Delta W > 0$, which blocks gradients for negative weights. Is this a typo or an intentional design choice?__
>
> Yes, it was a typo and it should be $| \Delta W | > 0$. We have corrected __Equation 3__ in the revised manuscript and appreciate the reviewer for pointing this out.
>
> > __Q3. Definition of sparsity__
>
> We apologize for the confusion. The ‘total sparsity’ in __Table 1__ refers to the fraction of parameters that remain unchanged (i.e., un-updated) in the entire model. We have clarified this definition in the revised manuscript __(Lines 321-323)__.
>
> __Reference__
>
> [1] Louizos et al., “Learning Sparse Neural Networks through L0 Regularization.” ICLR (2018).
>
> [2] Kusupati et al., “Soft threshold weight reparameterization for learnable sparsity.” ICML (2020).
>
> [3] Batatia et al., “MACE: Higher Order Equivariant Message Passing Neural Networks for Fast and Accurate Force Fields.” NeurIPS (2022).
>
> [4] https://github.com/ACEsuit/mace/discussions/165
>
> [5] Kovács et al., “MACE-OFF: Short-Range Transferable Machine Learning Force Fields for Organic Molecules.” Journal of the American Chemical Society, 147, 21, 17598–17611 (2025).
>
> [6] Batatia et al., “A foundation model for atomistic materials chemistry.” The Journal of Chemical Physics, 163, 184110 (2025).
>
> [7] Wang et al., “ELoRA: Low-Rank Adaptation for Equivariant GNNs.” ICML (2025).

---

### Official Review · Reviewer_1j46 · 2025-11-01

**Soundness:** 3
**Presentation:** 3
**Contribution:** 3
**Rating:** 6
**Confidence:** 3

**Summary:**

The paper proposes sparsity-promoting fine-tuning for E(3)-equivariant architectures. The sparsity is achieved by introducing learnable parameters tau that control sparsity. The method is applied to the MACE architecture and benchmarked on the Inorganic crystals, Revised MD17, TM-O-Spin, and MP-mag datasets. For these datasets, the accuracies of the non-fine-tuned model, the model fitted from scratch, the fully fine-tuned model, the model fine-tuned by ELoRA, and the model fine-tuned by the proposed methods with two sparsity levels are presented. Additionally, the applicability of the presented approach for model's interpretation was studied.

**Strengths:**

The method produces the most accurate models compared to other approaches most of the time. Furthermore, the method is competitive with very high reported sparsity.

The benchmarking setups are diverse, including small molecules, crystals, and extension to other targets over energies and forces, along with magnetic degrees of freedom.

The method helps with the interpretability of fine-tuned models.

**Weaknesses:**

It seems that the method doesn't reduce computational cost and memory requirements during finetuning as the achieved sparsity would suggest. During training, it maintains the coefficients tau, which, to the best of my understanding, mirror each of the dense parameters. So, their total number equals the number of parameters in all layers that undergo finetuning.

The equivariance is preserved not by incorporating the equivariance constraints into the finetuning method itself, but instead by applying it only to scalar path weights of the model. In other words, any finetuning technique applied to these layers would preserve equivariants. Therefore, I don't think that the positioning of the method for equivariant models, e.g., in the title of the paper, is fully justified.

**Questions:**

Is it correct that the memory requirements match those of the full finetuning because of the need to store dense tensors of the tau coefficients?

Could you provide practical finetuning times and memory requirements for full finetuning, ELoRA, and your method for the reported experiments?

Is it correct that the method is also equally applicable for invariant and unconstrained architectures?

---

> ### Author Response · Authors · 2025-11-24
> **Response 1/2**
>
> We sincerely appreciate reviewer 1j46’s constructive feedback. In response to the reviewer’s valuable comments, we have provided detailed answers accordingly.
>
> > __W1-1. Is $\tau$ mirror each of the dense parameters?__
>
> We would like to clarify that $\tau$ represents a single scalar, which is a layer-wise learnable threshold, and not equal to the size of each dense parameter. In other words, we allocate only a single scalar $\tau$ per each equivariant layer. We conjecture that the reviewer might have confused $\Delta W$ with $\tau$.
>
> > __W1-2. “So, their total number equals the number of parameters in all layers that undergo finetuning.”__
>
> Following from the above clarification (W1-1), $\Delta W$ mirrors the original dense parameters $W$ in our method, not $\tau$. Since $\tau$ is a single scalar per layer, the additional memory overhead from $\tau$ is negligible. Furthermore, we do not fine-tune the entire model but only target a selected subset of equivariant layers; therefore, the number of learnable parameters during fine-tuning is smaller than that of the full fine-tuning.
>
> > __W1-3. “It seems that the method doesn't reduce computational cost and memory requirements during finetuning …”__
>
> The reviewer's observation is correct, but we would like to clarify the scope of our contributions. Our method does not aim to reduce training-time computational cost or memory usage, and we do not claim such benefits in the paper. The primary purpose of our work is to induce sparsity in $\Delta W$, thereby improving adaptation accuracy and providing interpretability as demonstrated in __Figure 2__. In the fine-tuning scenario of our work, as the original parameters $W$ remain dense, inducing sparsity only in $\Delta W$ does not inherently reduce computational cost. We have clarified that the objective of our method is not to reduce computational costs in the revised manuscript __(Lines 172-175)__.
>
> > __Q1 & Q2. Practical fine-tuning times and memory requirements for each method__
>
> As we clarified in our response W1-1 through W1-3, $\tau$ is a single scalar parameter per layer, not a dense parameter; therefore, the memory overhead from $\tau$ is negligible. The primary memory consumption in our method instead comes from storing $\Delta W$ and the associated values, such as intermediate activations, gradients and optimizer states.
>
> In response to the reviewer’s request, we measured the training iteration time and memory consumption of full fine-tuning, ELoRA, and our method. Following our experimental setup, we fine-tuned the MACE-OFF23-medium model on the rMD17 Aspirin dataset and measured the average iteration time and memory usage, on a single NVIDIA H100 GPU. We performed 20 warm-up iterations and measured the average over the following 20 iterations, repeated three times, with our default batch size (64).
>
> |     Method    | Avg. time (s) | Memory (MB) |
> |:-------------:|:-------------:|:-----------:|
> | Full          |    0.14492    |    5291.7   |
> | ELoRA (r=16)  |    0.16556    |    5805.2   |
> | Ours (Linear) |    0.15949    |    5302.0   |
> | Ours (FCTP)   |    0.14977    |    5828.3   |
> | Ours (All)    |    0.16175    |    5834.3   |
>
> __Training time__: Our methods exhibit a small overhead (typically 4-7 % slower than full fine-tuning). Notably, all our methods are faster than ELoRA. This is because the matrix multiplication required in ELoRA to reconstruct $\Delta W$ becomes a bottleneck in the GPU.
>
> __Memory usage__: The memory overhead of our method when tuning only equivariant linear layers (‘Linear’ setting) is marginal compared to full fine-tuning. When all equivariant layers are fine-tuned using our method (‘All’ setting), the additional memory cost is comparable to that of ELoRA. Unlike LLMs, MLIPs typically do not benefit from significant memory reductions because memory usage is dominated by the intermediate activations and gradients required for autograd-based force prediction, rather than by the optimizer state size.
>
> __Convergence speed__: We further analyzed the overall convergence speed. By plotting the total loss against training steps on the rMD17 Aspirin and LAM AgAu datasets (see __Appendix Figure II__), we observed that our method outperforms ELoRA in convergence speed and is slightly faster than the full fine-tuning scenario.
>
> We have added __Section 5.5__ to the revised manuscript to include the above discussion.
>
> __(Continued in the following response)__

---

> ### Author Response · Authors · 2025-11-24
> **Response 2/2**
>
> > __W2-1. “... applying it only to scalar path weights of the model”__
>
> We respectfully clarify that __‘scalar path weights’ does not mean our method updates only scalar features ($\ell=0$)__, but also updates all tensor features. In equivariant MLIPs, each feature transformation, including tensor features with $\ell>0$, is realized through pre-defined Clebsch-Gordan tensor products, as described in __Appendix A (Equation 5)__. Crucially, each of these Clebsch-Gordan paths is associated with a learnable parameter that is scalar-valued. In other words, our method adjusts the scalar coefficients that scale Clebsch-Gordan interactions across all $\ell$ channels, while leaving the underlying equivariant tensor structure intact.
>
> In the revised manuscript, we have clarified this point in __Section 3.1 (Lines 168-169)__.
>
>
> > __W2-2. “I don't think that the positioning of the method for equivariant models … is fully justified”__
>
> We appreciate the reviewer's constructive comment. We acknowledge that our method operates on top of pre-existing equivariant frameworks, and we do not claim to introduce a novel mechanism to enforce equivariance. However, we emphasize that preserving equivariance during fine-tuning remains as a critical requirement. Given that equivariant architectures have become standard in the MLIP domain, __the positioning of our work specifically for equivariant models is justified by the unique synergy that emerges when sparsity is integrated with this framework.__
>
> Unlike generic neural network models, the learnable parameters in equivariant MLIPs directly correspond to physical interaction paths. Therefore, applying sparsity in this context offers the distinct advantage of physical interpretability, as shown in __Figure 2__, which is one of our core contributions.
>
> We have revised the manuscript at __Lines 65 and 73__ to explicitly clarify this positioning.
>
> > __Q3. Is it correct that the method is also equally applicable for invariant and unconstrained architectures?__
>
> Yes, our sparsity-promoting fine-tuning is equally applicable to general neural network architectures, including invariant or unconstrained ones. The notion of sparsity itself is independent of the presence of equivariance or architectural constraints. Nevertheless, as explained in W2-2, we focus on equivariant MLIPs in this work because they uniquely benefit from sparsity-driven path-level interpretability, and also because most recent materials foundation models adopt equivariant architectures.

---

### Author Response · Authors · 2025-11-24
**General Response**

We sincerely thank all four reviewers for their time and valuable comments.

The reviewers highlighted the strengths of our work as follows:
* _“The work addresses an important and timely topic.”_ (Reviewer __cUD3__)
* The method achieves _“the most accurate models compared to other approaches”_ with _“few updated parameters,”_ outperforming full fine-tuning and ELoRA at both low and high sparsity scenarios (Reviewer __1j46__, __KPDY__, __bcJ7__, __cUD3__)
* _"Diverse_ and _rigorous and well-motivated"_ experimental design with _"appropriate baselines"_ including extension to magnetic moment prediction (Reviewer __1j46__, __KPDY__, __cUD3__)
* The method is _"simple, and the implementation appears lightweight."_ (Reviewer __bcJ7__)
* The method _"helps with the interpretability of fine-tuned models"_ and provides novel analysis of sparse update patterns (Reviewer __1j46__, __KPDY__)

On the other hand, the reviewers raised the common questions and concerns:
* About the computational efficiency (Reviewer __1j46__, __KPDY__, __bcJ7__, __cUD3__)
* Discrepancy between our paper and the original MACE paper [1] (Reviewer __KPDY__, __bcJ7__)
* $\Delta W$ and equivariance (Reviewer __1j46__, __cUD3__)

In the point-by-point responses to each reviewer, we provide detailed answers to the key questions above, along with responses to individual comments and several clarifications. In addition, we have uploaded the revised manuscript to reflect the content addressed in our responses.

All modifications and additions in the updated manuscript are highlighted in pink, with the major revisions as follows:
* Clarified the definition of sparsity in __Section 5.1__ (Reviewer __KPDY__)
* Changed the notation for sparsity from 'S' to 'Sp.' in __Table 1__ (Reviewer __cUD3__)
* Corrected the erroneously reported value of __48.75__ to __70.46__ in __Table 2__ (Reviewer __KPDY__)
* Added an analysis of ELoRA’s parameter update patterns in __Section 5.4__ (Reviewer __KPDY__)
* Added experiments and analysis on computational cost in __Section 5.5__, and the convergence speed analysis of each method in __Appendix D__ and __Appendix Figure II__. (Reviewer __1j46__, __KPDY__, __bcJ7__, __cUD3__)
* Added experimental results on the NequIP architecture [2] in __Appendix D__ and __Appendix Table III__ (Reviewer __bcJ7__)

We thank the reviewers once again for their thorough review of our work and look forward to any further feedback.

__Reference__

[1] Batatia et al., “MACE: Higher Order Equivariant Message Passing Neural Networks for Fast and Accurate Force Fields.” NeurIPS (2022).

[2] Batzner et al., “E(3)-equivariant graph neural networks for data-efficient and accurate interatomic potentials.” Nature Communications 13, 1, 2453 (2022).

---

### Author Response · Authors · 2025-12-03
**Summary of our Rebuttal**

We sincerely appreciate all reviewers for their constructive feedback. As detailed in our general response and individual replies, we have revised the manuscript and addressed each comment thoroughly. To facilitate the final evaluation, we provide below a concise summary of the key concerns raised across reviews and our corresponding responses.


> __The reported MACE results do not match with those in the original paper. (KPDY, bcJ7)__

This is due to architectural differences. The original MACE paper adopts a significantly larger backbone model than ours. Our experiments are based on de facto standard foundation models (MACE-OFF23, MACE-MP-0b3) for fair and consistent comparison with the ELoRA baseline (Sec. 4.2).

> __Compare with NequIP-OAM-L to demonstrate the effectiveness of the sparse finetuning method. (bcJ7)__

Following the suggestion, we additionally compare with NequIP-OAM-L in Table III, Appendix D. It indicates that our method generalizes across equivariant architectures and mostly surpasses full fine-tuning.

> __Does the proposed method update only scalar path weights to preserve equivariance? (1j46, cUD3)__

No, we correct the reviewers’ misunderstanding. Our method updates all tensor path weights, not just the scalars. We revised Sec. 3.1 (Lines 168-169) to make this clearer.

> __The proposed method seems to be computationally less efficient than existing methods. (KPDY, bcJ7, 1j46, cUD3)__

We clarify that our method does not primarily aim for computational savings; rather, we mainly aim to induce sparsity in the weight updates $\Delta W$ to provide physical interpretability (Fig. 2). To address this concern, we report all types of computational cost (Sec. 5.5), indicating that the additional training cost is lower than ELoRA. Inference cost is identical to the original foundation model.

> __Clarify the novelty and contribution. (KPDY)__

We demonstrate that the naive Soft Threshold Weight Reparametrization is unsuitable for equivariant settings, while our decoupled formulation enables stable sparsification and interpretable interaction path isolation for the first time, which full-fine-tuning and existing methods (e.g., ELoRA) cannot provide.

---

### Meta-Review · Area_Chair_cPp1 · 2025-12-24

**Summary:**

The paper proposes an equivariant version of Soft Threshold Reparameterization (STR) that can be applied directly to geometric deep learning models, such as machine learning interatomic potentials (MLIPs). The method is first motivated and described by the authors with the main results showing comparable performance to other fine-tuning methods. The post-rebuttal version of the papers includes additional models (Nequip) as well as a computational performance analysis of the method, both of which show comparable performance to the baselines studied. The main distinction emphasized by the authors is that their method provides additional interpretability compared to other fine-tuning methods that do not emphasize sparse fine-tuning.

The reviewers generally positively note the strength of the benchmarks presented as well as the applicability of the method, which can be applied to a diverse set of models. The main weaknesses pointed out by the reviewers include a lack of computational benchmarking, which was included in the revised version, along with clarifications on the models used for benchmarking, which is also included in the rebuttal and revised version. Some reviewers also mention novelty as a concern along with the interpretability argument requiring further strength. Given that this is the main distinguishing feature highlighted by the authors, this aspect is important. An additional contribution of the paper is the TM-O-Spin dataset, which is calculated using open-source DFT.

**Reviewer Concerns:**

* Reviewer 1j46's concerns seem mostly addressed.
* Reviewer KPDY's concerns seem somewhat addressed. The rebuttal and revised version include many of the requested clarifications and additional data on computational efficiency. The concern about novelty and making use of sparsity for interpretability are discussed but could have been strengthened with additional evidence (e.g. analysis from rMD17).
* Reviewer bcJ7's concerns appear mostly addressed with clarifications on the results and additional studies of the Nequip architecture.
* Reviewer cUD3's concerns appear mostly addressed.

**Reviewer Scores:**

* Reviewer 1j46: keep score
* Reviewer KPDY: keep score
* Reviewer bcJ7: raise score to 6.
* Reviewer cUD3: raise score to 6.

---

### Decision · Program_Chairs · 2026-01-26

Accept (Poster)